# UFT: Unifying Supervised and Reinforcement Fine-Tuning

**Mingyang Liu, Gabriele Farina & Asuman Ozdaglar**
LIDS, EECS
Massachusetts Institute of Technology
Cambridge, MA 02139, USA
{liumy19,gfarina,asuman}@mit.edu

## Abstract

Post-training has demonstrated its importance in enhancing the reasoning capabilities of large language models (LLMs). The primary post-training methods can be categorized into supervised fine-tuning (SFT) and reinforcement fine-tuning (RFT). SFT is efficient and well-suited for small language models, but it may lead to overfitting and limit the reasoning abilities of larger models. In contrast, RFT generally yields better generalization but depends heavily on the strength of the base model. To address the limitations of SFT and RFT, we propose Unified Fine-Tuning (UFT), a novel post-training paradigm that unifies SFT and RFT into a single, integrated process. UFT enables the model to effectively explore solutions while incorporating informative supervision signals, bridging the gap between memorizing and thinking underlying existing methods. Notably, UFT outperforms both SFT and RFT in general, regardless of model sizes. Furthermore, we theoretically prove that UFT breaks RFT's inherent exponential sample complexity bottleneck, showing for the first time that unified training can exponentially accelerate convergence on long-horizon reasoning tasks.

## 1 Introduction

When humans learn a new subject, we typically practice with problem sets (thinking) and try to understand the solutions when we encounter difficulties (memorizing). There are also counterparts in fine-tuning LLMs, which is

- **Supervised Fine-Tuning (SFT).** Memorizing the collected reasoning trace (solution) by maximizing the log-likelihood of it.
- **Reinforcement Fine-Tuning (RFT).** Exploring the reasoning space of LLM and improving the performance according to the signal from a verifier of the *final answer* (thinking).

However, unlike humans, learning and thinking are disentangled during the training of language models. Specifically, prior work [DeepSeek-AI et al., 2025, Zhou et al., 2023, Muennighoff et al., 2025, Liu et al., 2025, Zeng et al., 2025] typically applies either SFT or RFT throughout the fine-tuning phase, or applies RFT only after SFT completes (cf. Figure 1). The choice of the proper fine-tuning algorithm depends on the LLM's capacity and the task's complexity. Specifically, when the LLM is weak, SFT typically works better since the LLM cannot explore the correct answer during reinforcement learning [Pan et al., 2025], due to the sparse reward caused by the verifier-based reward model. On the other hand, when the LLM is strong, RFT generalizes better [Xie et al., 2025, Chu et al., 2025].

---

[1]The source code is available at https://github.com/liumy2010/UFT.

39th Conference on Neural Information Processing Systems (NeurIPS 2025).

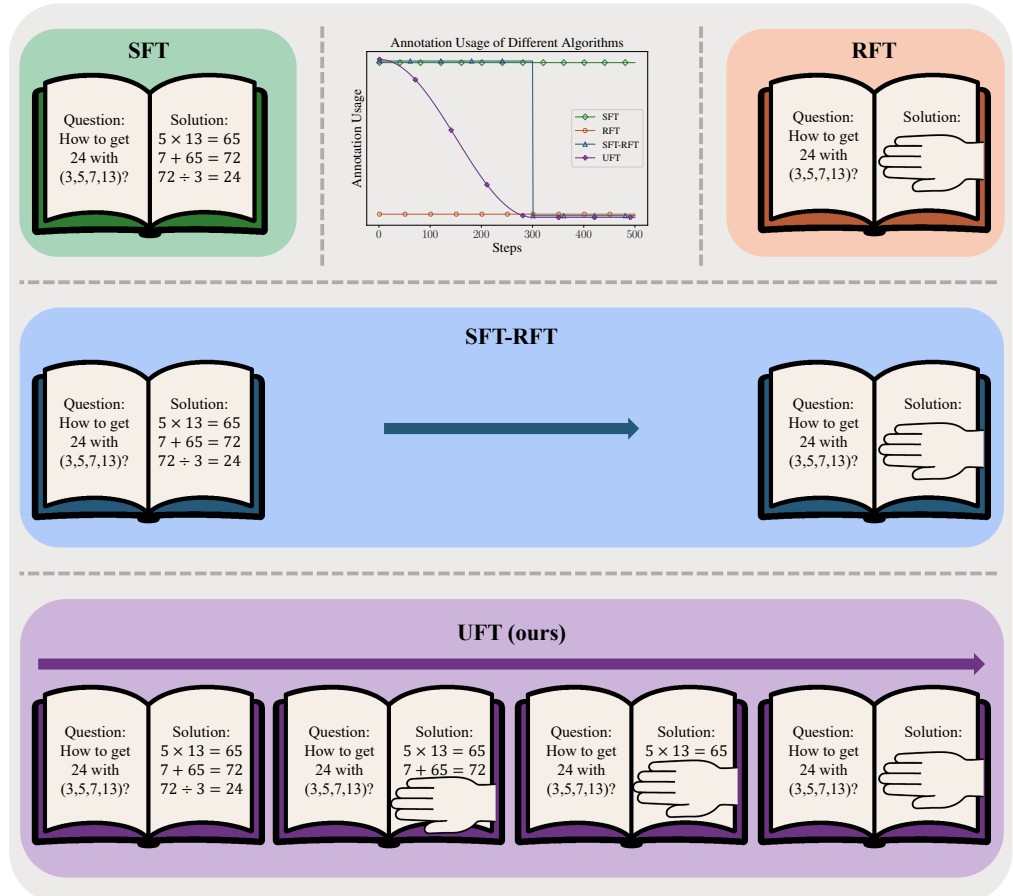

Figure 1: (top left, top right, middle, bottom). The illustration of SFT, RFT, SFT-RFT, and UFT, respectively. SFT-RFT refers to applying RFT after an initial SFT stage [DeepSeek-AI et al., 2025, Zeng et al., 2025]. (Top, center). shows the annotation usage of different algorithms over training. Curves are slightly shifted for better visibility.

To get the best of both worlds, we propose Unified Fine-Tuning (UFT), which unifies SFT and RFT and enriches the reinforcement learning signal with supervised feedback, enabling the model to acquire new knowledge during fine-tuning more efficiently. In Figure 1, SFT-RFT refers to the common practice of initiating reinforcement learning from a supervised fine-tuned model, as widely adopted in the literature [DeepSeek-AI et al., 2025, Zeng et al., 2025]. As shown in Figure 1 (top left), SFT uses full annotations (solutions) throughout training, whereas RFT does not use any annotations at all (Figure 1, top right). Similarly, SFT-RFT begins with SFT using full annotations, but once the RFT phase starts, it discards all annotations and relies entirely on exploration. In contrast, our method, UFT, offers a smooth transition from SFT to RFT, preserving the annotation signal early on and gradually reducing it as the model becomes capable of self-guided reasoning.

The most relevant work to UFT is Learning Reasoning through Reverse Curriculum Reinforcement Learning ($R^3$) [Xi et al., 2024], which proposes a curriculum learning method that concatenates the problem with a slice of the solution (hint, cf. Figure 4 left). While $R^3$ treats hints primarily as exploration aids, UFT further integrates them as part of the supervision signal. This unification enables reinforcement learning not just to search, but to learn from existing solutions, effectively raising the performance ceiling imposed by the model's pretraining capacity (cf. Figure 2). A detailed comparison with related work is postponed to Appendix A.

Figure 2 shows the accuracy of different algorithms over time, while the training set is Countdown [Wikipedia contributors, 2025, Pan et al., 2025], MATH(3,4,5) (levels 3–5 only) [Hendrycks et al., 2021, Zeng et al., 2025], and the Knights and Knaves logic puzzle (Logic) [Xie et al., 2025]. Base

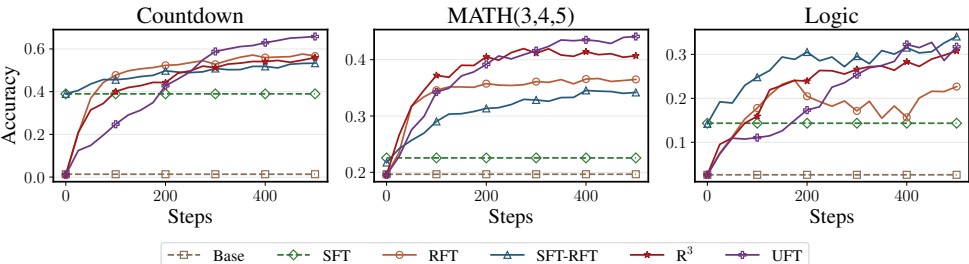

Figure 2: Presentation for different algorithms' accuracy when trained on Countdown [Wikipedia contributors, 2025], MATH(3,4,5) (level 3-5 only) [Hendrycks et al., 2021, Zeng et al., 2025], and the Knights and Knaves logic puzzle (Logic) [Xie et al., 2025]. Accuracy is averaged over Qwen2.5 models of sizes 0.5B, 1.5B, and 3B [Qwen et al., 2025]. Base refers to the model without fine-tuning, and $R^3$ is the curriculum reinforcement learning baseline [Xi et al., 2024]. The figure shows that UFT outperforms both SFT and RFT, while the relative performance of SFT and RFT varies depending on task complexity.

refers to the model before fine-tuning, and $R^3$ represents the curriculum reinforcement learning baseline [Xi et al., 2024]. As shown in the figure, UFT generally outperforms all other algorithms. Furthermore, we provide the evaluation on various benchmarks, and the results are shown in Table 8.

Moreover, we theoretically prove that RFT [DeepSeek-AI et al., 2025, Zeng et al., 2025, Liu et al., 2025] suffers from an inherent sample complexity bottleneck, which is exponential in the length of the reasoning. In contrast, the unified training paradigm in UFT can improve the sample complexity to a polynomial dependence on the reasoning length, which is an exponential improvement over RFT.

## 1.1 Contribution

We state the contribution of this paper in the following.

1. **Integration of Supervision and Reward Signal.** UFT provides a general framework that integrates the supervision from SFT and reward from RFT into a single training paradigm. UFT blends reward optimization with log-likelihood maximization on hints (partial solution), and smoothly transitions from fully supervised to fully reinforcement learning. Such integration allows models to explore and learn simultaneously, addressing the trade-off between memorization (SFT) and generalization (RFT) in a principled way.

2. **Theoretical Justification.** We provide a theoretical analysis of UFT, proving it achieves polynomial sample complexity dependence on reasoning length, compared to the exponential complexity required by standard RFT. This result formally establishes the efficiency gains from unifying learning (cf. Section 4).

3. **Empirical Validation Across Model Scales and Tasks.** We evaluate the algorithms by training Qwen2.5-0.5/1.5/3B [Qwen et al., 2025] and Llama3.2-1/3B [Grattafiori et al., 2024] on Countdown [Wikipedia contributors, 2025, Pan et al., 2025], MATH [Hendrycks et al., 2021], and the Knights and Knaves logic puzzle (Logic) [Xie et al., 2025]. UFT consistently outperforms previous methods, showing robustness across domains and models (cf. Section 5).

## 2 Preliminaries

**Notation.** For any integer $n > 0$, let $[n] := \{1, 2, \cdots, n\}$ and $\Delta^n := \left\{ \boldsymbol{x} \in [0,1]^n : \sum_{i=1}^{n} x_i = 1 \right\}$ be the $n-1$-dimenional probability simplex. For any two distribution $\boldsymbol{x}, \boldsymbol{y} \in \Delta^n$, let $\mathrm{KL}\left(\boldsymbol{x} \| \boldsymbol{y}\right) := \sum_{i=1}^{n} x_i \log \frac{x_i}{y_i}$ denote the KL-divergence between $\boldsymbol{x}$ and $\boldsymbol{y}$. For any discrete set $\mathcal{S}$, let $|\mathcal{S}|$ be its cardinality.

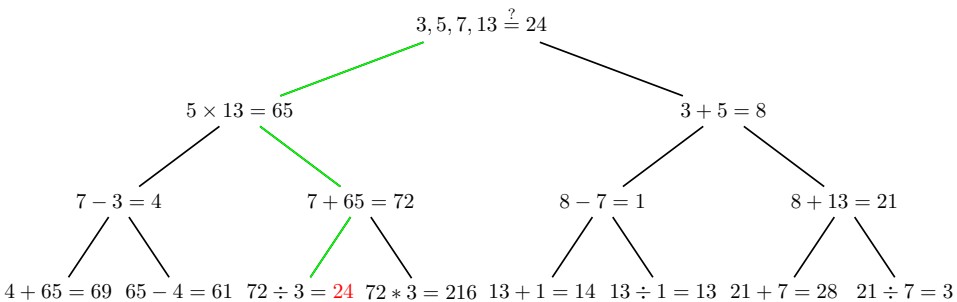

Figure 3: An illustration of the Countdown game, where the goal is to obtain 24 by applying basic arithmetic operations $(+, -, \times, \div)$ to the numbers $(3, 5, 7, 13)$. The green path represents the correct solution.

**Search Tree.** The problem-solving process can be represented as a *search tree*, as illustrated in Figure 3. Except for the leaf nodes, each node (also referred to as a *state*—we use the terms node and state interchangeably) in the search tree has $B$ children, where $B$ is the *branching factor*. Each child represents a different next token (or next sentence) to be generated, so a path from the root to a leaf node corresponds to a complete solution to the problem. The tree has a height of $H$, with the root at height 0, and each node's height equal to its parent's height plus one.

Let $\mathcal{S}_h$ denote the set of nodes with height $h \in \{0, 1, \cdots, H\}$ and $\mathcal{S} := \bigcup_{h=0}^{H} \mathcal{S}_h$. Note that $|\mathcal{S}_0| = 1$ since it only contains the root $s_{\text{root}}$, and $|\mathcal{S}_{h+1}| = B \cdot |\mathcal{S}_h|$. Therefore, there are $\sum_{h=0}^{H} B^h = \frac{B^{H+1}-1}{B-1}$ nodes in total. Once reaching a leaf node $s \in \mathcal{S}_H$, the model will receive reward $\mathcal{R}(s) \in [0, 1]$. A policy can be written as $\pi \colon \bigcup_{h=0}^{H-1} \mathcal{S}_h \to \Delta^B$, where $\pi(a \mid s)$ is the probability of selecting the $a^{th}$ child of $s$. For any state-action pairs $(s, a) \in \mathcal{S} \times [B]$, let $\mathcal{T}(s, a) \in \mathcal{S}$ be the child at the branch $a$ of state $s$, and $\mathcal{T}(s, a) = \emptyset$ for $s \in \mathcal{S}_H$. The value function of policy $\pi$ is written as $V^\pi \colon \mathcal{S} \to [0, 1]$. We write $s_{h_0} = s, (s_h)_{h=h_0}^{H} \sim \pi$ as the trajectory starting from $s$ and sampled according to $\pi$, *i.e.*, $a_h \sim \pi(\cdot \mid s_h), s_{h+1} = \mathcal{T}(s_h, a_h)$. For any $h_0 \in \{0, 1, \cdots, H\}$ and $s \in \mathcal{S}_{h_0}$, we define $V^\pi(s) := \mathbb{E}_{s_{h_0}=s,(s_h)_{h=h_0}^{H} \sim \pi} [\mathcal{R}(s_H)]$, which is the expected reward obtained by following policy $\pi$ starting from node $s$.

Let $\pi^* \in \arg\max_\pi V^\pi(s_{\text{root}})$ denote the optimal (deterministic) policy that achieves the highest expected reward[1]. Let $V^* := V^{\pi^*}(s_{\text{root}})$ be the expected reward of the optimal policy $\pi^*$. Since $\pi^*$ is deterministic, let $(s_0^*, a_0^*, s_1^*, a_1^*, \cdots, s_H^*)$ represent the path from the root to a leaf node by following $\pi^*$, where $s_0^* = s_{\text{root}}$.

# 3 Unified Fine-Tuning (UFT)

In this section, we introduce the two key features of UFT: (i) an exploration mechanism guided by hint, which improves sample efficiency by mitigating the sparse reward problem common in rule-based reinforcement learning [DeepSeek-AI et al., 2025]; and (ii) a hybrid training objective that combines reinforcement learning with a log-likelihood term on hints, which provides a more informative learning signal and enables the model to acquire knowledge more effectively during fine-tuning.

## 3.1 Exploration with Hint

Although RFT is beneficial for training large models [DeepSeek-AI et al., 2025], several recent studies [Pan et al., 2025] report that small models often fail to reason effectively, as they may never explore the correct answer even once due to the sparse reward. Additionally, other work has found that RFT's final performance is constrained by base models' capabilities [Gandhi et al., 2025].

---

[1]There exists at least one deterministic optimal policy, and we choose such a policy as $\pi^*$.

To address the sparse reward issue, UFT guides exploration using a hint, that is, trajectory sampling starts from the concatenation of the problem description and a hint, which is a partial solution to the problem (cf. Figure 4). In this way, models will explore the correct answer more frequently.

RFT can be modeled as the task of finding a path from the root of the problem-solving tree to a leaf node that represents the correct answer. As shown in Figure 3, RFT needs to identify the green path. However, the problem-solving tree for real-world tasks, such as math problems, typically contains an enormous number of nodes, making it difficult for an LLM to discover the correct path through exploration alone. To make matters worse, under the rule-based reward model proposed in DeepSeek-AI et al. [2025], only a small fraction of the leaf nodes correspond to correct answers, resulting in the well-known sparse reward problem [Ladosz et al., 2022].

We address this challenge by concatenating the problem with a partial solution, referred to as the *hint*, to guide the model towards the correct answer. Figure 4 (left) provides an example of UFT's prompt.

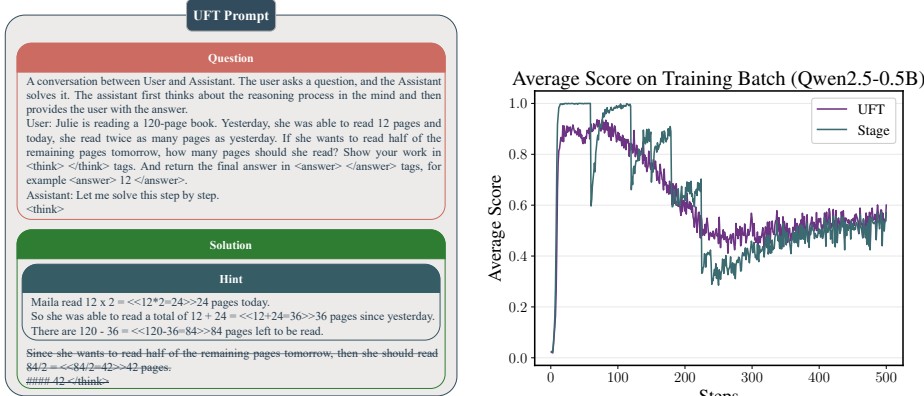

Figure 4: (left). An illustration of the UFT prompt. We adopt the prompting template from TinyZero [Pan et al., 2025], which is similar to that used in Deepseek-R1 [DeepSeek-AI et al., 2025]. The hint consists of a slice of the full solution. During training, the question prompt and the hint are concatenated and fed to the model. (right). An illustration of the training curve of Qwen2.5-0.5B. Stage and UFT keep zero hint since step 300.

### 3.1.1  Hint Length Sampling

Since we are ultimately interested in the LLM's performance when the hint length is zero, the hint must be gradually shortened during training. A natural idea is to subtract the hint length by a constant amount regularly, which is referred to as the staged reinforcement learning [Xi et al., 2024]. However, because solutions typically consist of no more than 10 sentences, changes in hint length can cause a significant distribution shift, leading to unstable training (cf. Figure 4, right).

To avoid distribution shift during training, Xi et al. [2024] samples the hint length uniformly from all possible values throughout the training. However, relying on hints throughout training introduces a significant distribution mismatch between training and evaluation. This often leads to performance collapse at test time, where no hints are available. To address this, UFT employs a smoothed reduction of hint length to zero, which (i) avoids drastic distribution shifts and (ii) better aligns the training distribution with the evaluation distribution.

Specifically, we maintain a variable $p \in [0, 1]$, representing the proportion of the solution revealed to the LLM as a hint. The value of $p$ gradually descends during training according to cosine annealing (cf. (B.1)) [Loshchilov and Hutter, 2017]. Let $l$ be the random variable indicating the hint length, and let $L$ be the total length of the solution (*e.g.*, number of sentences). By definition, we require $l \in \{0, 1, \cdots, L\}$ and $\mathbb{E}[l] = p \cdot L$, so that the expected hint length matches the proportion $p$. To achieve this, we sample $l \sim \text{Binomial}(L, p)$ from a binomial distribution[2]. It is straightforward to

---

[2]$\Pr(l = l_0) = \binom{L}{l_0} p^{l_0} (1 - p)^{L - l_0}$ for any $l_0 \in \{0, 1, \cdots, L\}$. In other words, $l$ is the number of heads obtained when tossing $L$ independent coins, each landing heads with probability $p$.

verify that $\mathbb{E}[l] = L \cdot \mathbb{E}[c_1] = p \cdot L$. In Appendix B.3, we further show that UFT is robust to the choice of hint length distribution, and we choose binomial distribution due to its simplicity.

Compared to stage-wise hint length reduction, UFT provides a smoother transition from long to short hints. The training curves of these algorithms are shown in Figure 4 (right). We can see that the training curve of UFT is smoother and converges faster than that of the staged reinforcement learning. Note that staged reinforcement learning and UFT do not use any hint since step 300.

As shown in Figure 5, although RFT (cosine), which is RFT equipped with the cosine annealing hint length scheduler, outperforms $R^3$ (uniform sampling), it is still worse than SFT-RFT. Furthermore, for Llama-3.2-1B, RFT (cosine) is even worse than SFT alone. This implies that the model's performance is hindered by its knowledge gained through pretraining [Gandhi et al., 2025], which motivates the second modification of UFT introduced in Section 3.2, an additional log-likelihood term in the objective function.

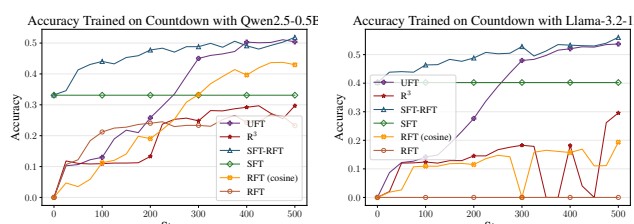

Figure 5: An ablation study of different hint length schedulers. RFT (cosine) refers to reinforcement learning with our cosine annealing hint length scheduler proposed in this section.

## 3.2 Objective Function Modification

The hinted RFT only enables LLMs to explore the correct solution more frequently, but remains inefficient at injecting new knowledge into the LLMs. This inefficiency arises because each sampled trajectory provides limited information, essentially a signal (correct/incorrect), which provides far less information than the supervision signal in SFT. In contrast, SFT enables more efficient knowledge acquisition, but suffers from poor generalization [Xie et al., 2025, Zeng et al., 2025]. To get the best of both worlds, UFT introduces an additional log-likelihood term to the objective function of RFT, allowing the model to learn from the informative supervision signal and still benefit from the generalization of RFT.

For notational simplicity, let $s_0 = s_{\text{root}}, (s_h, a_h)_{h=0}^{H-1} \sim \pi$ denote the shorthand for $a_h \sim \pi(\cdot \mid s_h)$ and $s_{h+1} = \mathcal{T}(s_h, a_h)$, i.e., $(s_h, a_h)_{h=0}^{H-1}$ represents a trajectory sampled according to $\pi$ starting at $s_{\text{root}}$. Formally, let $\mathcal{J}^{\text{value}}\left((s_h, a_h)_{h=0}^{H-1}\right)$ denote the objective function associated with the expected reward[3]. Then, let $\beta > 0$ be the hyperparameter controlling the KL divergence, we have

$$\mathcal{J}^{\text{RFT}} = \mathbb{E}_{s_0=s_{\text{root}},(s_h,a_h)_{h=0}^{H-1}\sim\pi}\left[\mathcal{J}^{\text{value}}\left((s_h, a_h)_{h=0}^{H-1}\right) - \beta\sum_{h=0}^{H-1}\text{KL}\left(\pi(\cdot\mid s_h)\|\pi^{\text{ref}}(\cdot\mid s_h)\right)\right] \quad (3.1)$$

$$\mathcal{J}^{\text{UFT}} = \mathbb{E}_{\substack{l,s_0=s_{\text{root}}\\(s_h,a_h)_{h=0}^{l-1}\sim\pi^*,\\(s_h,a_h)_{h=l}^{H-1}\sim\pi}}\left[\mathcal{J}^{\text{value}}\left((s_h, a_h)_{h=l}^{H-1}\right) - \beta\sum_{h=l}^{H-1}\text{KL}\left(\pi(\cdot\mid s_h)\|\pi^{\text{ref}}(\cdot\mid s_h)\right) - \beta\sum_{h=0}^{l-1}\text{KL}\left(\pi^*(\cdot\mid s_h)\|\pi(\cdot\mid s_h)\right)\right]$$

$$(3.2)$$

Compared to the objective function of GRPO, UFT adds an additional term $\beta\sum_{h=0}^{l-1}\text{KL}\left(\pi^*(\cdot\mid s_h)\|\pi(\cdot\mid s_h)\right)$, the KL divergence between the optimal policy and the current policy. Compared to $\mathcal{J}^{\text{value}}$, this term explicitly guides the policy towards optimality, and thus results in a faster convergence rate.

We remark that the optimal policy $\pi^*$ is unknown and we cannot compute $\beta\sum_{h=0}^{l-1}\text{KL}\left(\pi^*(\cdot\mid s_h)\|\pi(\cdot\mid s_h)\right)$ directly. However, thanks to the annotations contained in

---

[3]In GRPO [Shao et al., 2024], we have $\mathcal{J}^{\text{value}}\left((s_h, a_h)_{h=0}^{H-1}\right) := \frac{1}{H}\sum_{h'=0}^{H-1}\min\left\{\frac{\pi(a_{h'}\mid s_{h'})}{\pi^{\text{old}}(a_{h'}\mid s_{h'})}\widehat{A}_{h'},\right.$
$\left.\text{clip}\left(\frac{\pi(a_{h'}\mid s_{h'})}{\pi^{\text{old}}(a_{h'}\mid s_{h'})}, 1-\epsilon, 1+\epsilon\right)\widehat{A}_{h'}\right\}$, where $\pi$ is the current policy, $\pi^{\text{old}}$ is the policy at the previous step, $\widehat{A}_{h'}$ is the estimated advantage value in GRPO.

the dataset, we have access to a trajectory sampled according to $\pi^*$, *i.e.*, $(s_h^*, a_h^*)_{h=0}^{H-1} \sim \pi^*$, which can be used to estimate the KL-divergence. According to the definition of KL-divergence, minimizing $\mathrm{KL}\left(\pi^*(\cdot \mid s_h^*) \| \pi(\cdot \mid s_h^*)\right)$ is equivalent to minimizing $\sum_{a_h=1}^{B} \pi^*(a_h \mid s_h^*) \log \frac{1}{\pi(a_h \mid s_h^*)}$ (omit terms irrelevant to $\pi$), and $\log \frac{1}{\pi(a_h^* \mid s_h^*)}$ is an unbiased estimator of it, since $a_h^* \sim \pi^*(\cdot \mid s_h^*)$. Therefore, (3.2) can be equivalently written as

$$\mathcal{J}^{\mathrm{UFT}} = \mathbb{E}_{\substack{l, s_l = s_l^*, \\ (s_h, a_h)_{h=l}^{H-1} \sim \pi}} \left[ \mathcal{J}^{\mathrm{value}}\left((s_h, a_h)_{h=l}^{H-1}\right) - \beta \sum_{h=l}^{H-1} \mathrm{KL}\left(\pi(\cdot \mid s_h) \| \pi^{\mathrm{ref}}(\cdot \mid s_h)\right) + \beta \sum_{h=0}^{l-1} \log \pi(a_h^* \mid s_h^*) \right]. \quad (3.3)$$

Therefore, the UFT objective (3.3) can be interpreted as (i) maximizing the expected reward while (ii) staying close to the reference policy and (iii) memorizing the hint by maximizing the log-likelihood of producing the hint.

**Remark 3.1.** The name of Unified Fine-Tuning (UFT) comes from the fact that when $p \equiv 0$ for all steps during training, (3.3) is equivalent to RFT, since $\beta \sum_{h=0}^{l-1} \log \pi(a_h^* \mid s_h^*) = 0$. When $p \equiv 1$, then $\mathcal{J}^{\mathrm{value}}\left((s_h, a_h)_{h=l}^{H-1}\right) - \beta \sum_{h=l}^{H-1} \mathrm{KL}\left(\pi(\cdot \mid s_h) \| \pi^{\mathrm{ref}}(\cdot \mid s_h)\right) = 0$, so that (3.3) degenerates to SFT. An illustration can be found in Figure 1 (top middle).

It is noteworthy that after adopting the additional log-likelihood term, UFT's performance matches that of SFT-RFT for small models (cf. Figure 5). This suggests that UFT improves the ceiling of RFT by enabling the model to acquire new knowledge during post-training.

### 3.3 Algorithm Outline

Overall, the UFT algorithm proceeds as follows:

- Sample a batch of problems $\mathcal{B}$ along with their corresponding solutions.
- For each problem, sample a hint length $l$ according to Section 3.1.1.
- Concatenate each problem with its solution prefix of length $l$.
- Train the model on $\mathcal{D}$ using a reinforcement learning algorithm with the objective defined in (3.3).

The corresponding pseudocode is provided in Algorithm 1.

## 4 Theoretical Justification

In this section, we provide a theoretical justification for UFT. First, we show that the lower bound of RFT's sample complexity grows exponentially ($\mathcal{O}(B^H)$) as the tree height (reasoning length) increases. Second, we show that UFT may find the solution within a polynomial number of samples ($\mathcal{O}\left(BH^5 \log B\right)$), representing an *exponential* improvement of tree height $H$ in sample complexity.

Next, we define the sub-optimality gap in reward, which is the difference between the rewards for correct and incorrect solutions.

**Definition 4.1** (Sub-Optimality Gap)**.** There is a *sub-optimality gap* $\Delta > 0$ between the reward of optimal and suboptimal nodes. Formally, for any leaf node $s \in \mathcal{S}_H$ with reward $\mathcal{R}(s) < \max_{s' \in \mathcal{S}_H} \mathcal{R}(s')$, we have

$$\mathcal{R}(s) \leq \max_{s' \in \mathcal{S}_H} \mathcal{R}(s') - \Delta. \quad (4.1)$$

In this paper, there are only three possible outcomes for $\mathcal{R}(s)$, *i.e.*, no reward (incorrect format), format reward, and accuracy reward. Therefore, the sub-optimality gap

$$\Delta = (\text{accuracy reward}) - (\text{format reward}) = 1.0 - 0.1 = 0.9. \quad (4.2)$$

Next, we will give the lower bound on the RFT's sample complexity to achieve $50\%$ pass@1 success rate[4].

---

[4]The probability of reaching the correct answer when sampling a single trajectory.

**Theorem 4.2** (Lowerbound). For any integers $H \geq 1$, $B \geq 2$, and any RFT algorithm, there exists a problem with height $H$ and branching factor $B$, that satisfies the following: to achieve a $50\%$ pass@1 success rate, the algorithm needs to explore at least

$$\frac{B^H}{4} \tag{4.3}$$

nodes in $\mathcal{S}_H$. Moreover, when there are multiple nodes in $\mathcal{S}_H$ representing the correct solutions, *e.g.*, $K \geq 1$, any algorithm needs to explore at least $\frac{B^H}{4K}$ nodes in $\mathcal{S}_H$.

The proof constructs a set of problems with different correct solutions, which cannot be distinguished before exploring sufficient nodes in $\mathcal{S}_H$. The details can be found in Appendix C. Furthermore, the traditional lower bounds in reinforcement learning [Jin et al., 2018, Domingues et al., 2021] are built on the stochastic transitions of the Markov decision process, but the search tree's transition is deterministic, which requires a different construction.

Theorem 4.2 implies that when the reward is sparse, such as when $K$ is a constant, learning the optimal policy takes a number of iterations exponential in the height of the tree. This also justifies why long reasoning is generally difficult [Chai et al., 2025, Chen et al., 2025]. In the following, we will show that UFT *exponentially* improves the sample complexity. The full algorithm can be found in Algorithm 2.

**Theorem 4.3** (Informal). When $\beta$ is small enough, Algorithm 2 obtains a $50\%$ pass@1 success rate when the algorithm explores

$$\mathcal{O}\left(B\frac{H^5 \left(\log B\right)^2}{\Delta^2}\right) \tag{4.4}$$

nodes in $\mathcal{S}_H$.

The formal version is deferred to Appendix E. Note that the $50\%$ pass@1 in both Theorem 4.2 and Theorem 4.3 can be arbitrarily adjusted, and it only affects the sample complexity by a constant factor. From Theorem 4.3, we observe that the dependence on $H$ is reduced from $B^H$ to $H^5$, representing an exponential improvement enabled by the use of hints. Moreover, $\Delta^2$ in the denominator implies that the difference between accuracy reward and format reward should be large for fast convergence, which is also supported by empirical studies [Shao et al., 2024, Pan et al., 2025, Zeng et al., 2025].

## 5 Experiments

In this section, we present the experimental results of UFT. We demonstrate several key properties of UFT: (i) When the model is small ($\leq 1B$) and SFT outperforms RFT, UFT's performance matches that of SFT. (ii) When the model is large ($\sim 3B$) and RFT outperforms SFT due to better generalization, UFT's performance matches that of RFT (and sometimes even outperforms it, cf. Table 8).

In experiments, we train Qwen2.5-0.5B, Qwen2.5-1.5B, Qwen2.5-3B [Qwen et al., 2025], Llama-3.2-1B, and Llama-3.2-3B [Grattafiori et al., 2024] on Countdown [Wikipedia contributors, 2025, Pan et al., 2025], MATH(3,4,5) (only level 3-5 included) [Hendrycks et al., 2021, Zeng et al., 2025], and the Knights and Knaves logic puzzle (Logic) [Xie et al., 2025].

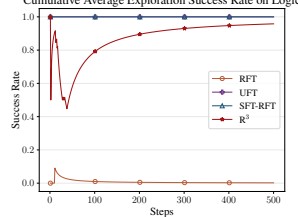

Figure 6: Qwen2.5-0.5B's cumulative average success rate for exploring the correct answer at each step when trained on Logic.

### 5.1 The Memorization of UFT

As shown in Figure 7, we can see that when the model is small, the improvement from RFT is marginal, since the model rarely explores the correct answer. As shown in Figure 6, when training Qwen2.5-0.5B on Logic, RFT rarely explores the correct answer, while UFT finds it at every single timestep.

Compared to R$^3$, where hints are also applied, UFT outperforms it since UFT (i) gradually shifts the distribution toward a hint length of zero, and (ii) maximizes the log-likelihood on hints to encode information about the solution in gradients. The proximity between the performance of UFT and SFT-RFT also supports the conclusion that UFT helps the model to memorize the solution when the model's initial capacity is not enough to solve it.

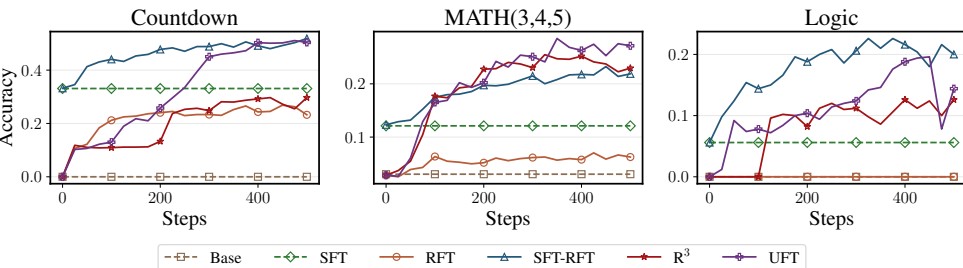

Figure 7: An illustration of the accuracy on the test dataset of Qwen2.5-0.5B. Base is the base model without fine-tuning. $R^3$ [Xi et al., 2024] trained the model with RFT and a uniform distribution over all hint lengths. SFT-RFT refers to training a supervised fine-tuned model with RFT, and UFT is our algorithm.

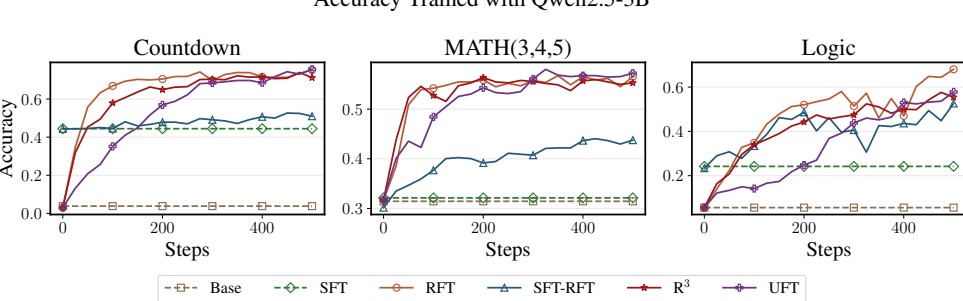

Figure 8: An illustration of the accuracy on test dataset of Qwen2.5-3B. Base refers to the base model without fine-tuning.

## 5.2 The Generalization of UFT

As shown in Figure 8, when the model is larger and its prior knowledge gained from pertaining is enough for reasoning, UFT generalizes well as RFT. In contrast, SFT and SFT-RFT are worse, since SFT leads to overfitting. These experiments show that UFT will automatically adapt to model size and enjoy the advantage of both SFT and RFT.

As shown in Figure 8, when the model is larger and its prior knowledge gained from pretraining is sufficient for reasoning, UFT generalizes well as RFT. In contrast, SFT and SFT-RFT perform worse, since SFT leads to overfitting. These experiments show that UFT automatically adapts to model size and benefits from the advantages of both SFT and RFT.

## 5.3 UFT Helps LLMs Learn New Knowledge

In Gandhi et al. [2025], it was found that Llama-3.2-3B's improvement through RFT is marginal compared to that of Qwen2.5-3B. This is because Llama gains less reasoning-related knowledge from pertaining, *e.g.*, backtracking and subgoal setting. In Figure 9, we can see that UFT significantly improves the performance of Llama-3.2. In Countdown, even Llama-3.2-1B outperforms Llama-3.2-3B fine-tuned by RFT after the same number of steps (250 steps). This supports the claim that UFT introduces new knowledge to the model, whereas RFT only helps the model utilize its existing knowledge [Yue et al., 2025].

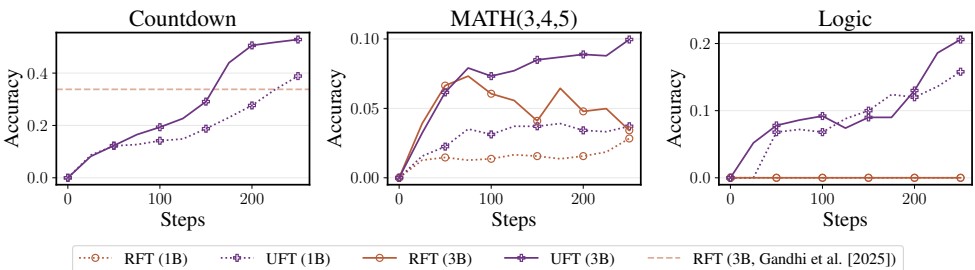

Figure 9: The comparison of Llama-3.2-1B/3B's behavior in Countdown/MATH/Logic when applying RFT/UFT. In Countdown, the dotted line is the accuracy of Llama-3.2-3B after 250 steps RFT reported in Gandhi et al. [2025] .

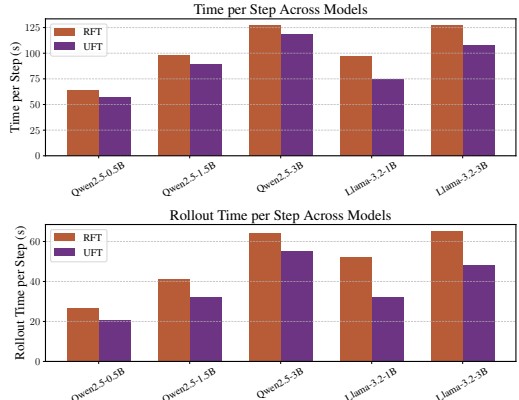

Figure 10: The computational cost per step of UFT and RFT.

## 5.4 Computational Cost of UFT

The computational costs of UFT and RFT are shown in Figure 10. Interestingly, UFT is faster than RFT, as it begins reasoning from a partial solution (a hint) rather than starting from scratch, thereby reducing the rollout cost during training.

## 6 Conclusion and Limitations

This paper proposes a novel fine-tuning framework, UFT, which unifies SFT and RFT. Empirically, we show that UFT outperforms both SFT and RFT in general. Specifically, by adopting UFT, small models tend to memorize while large models generalize. Theoretically, we prove that UFT achieves an exponential speed-up compared to RFT. However, throughout the paper, we use only the human-annotated solutions in the dataset and GRPO as the reinforcement learning algorithm. Moreover, we do not employ state-of-the-art (*e.g.*, 70B-scale) models in our experiments due to computational constraints, but provide evidence that UFT remains beneficial under these settings (see Appendix B.4). In the future, it would be interesting to explore the incorporation of advanced SFT and RFT techniques into UFT. For instance, using long chain-of-thoughts generated by large models [Muennighoff et al., 2025, Gandhi et al., 2025] for SFT, and choosing other reinforcement learning algorithms such as REINFORCE++ [Hu, 2025] and DAPO [Yu et al., 2025] as the reinforcement learning algorithm for UFT.

# 7 Acknowledgement

The authors would like to thank Jacob Andreas, Chanwoo Park, and Kaiqing Zhang for their valuable discussions. Mingyang Liu was supported by the Siebel Scholarship, and Gabriele Farina was supported in part by the CCF-2443068, ONR grant N000142512296, and an AI2050 Early Career Fellowship.

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

# A  Related Work

In this section, we introduce related work about SFT, RFT, and curriculum learning for reasoning.

**Supervised Fine-Tuning (SFT) for Reasoning.**   Different SFT methods for enhancing reasoning capability usually differ in the source of the collected reasoning trace. Zeng et al. [2025] uses traditional SFT, *i.e.*, learning from the human-annotated problem solutions. In contrast, Gandhi et al. [2025], Muennighoff et al. [2025] utilize long chain-of-thoughts solutions generated by some large models, such as Claude and Deepseek-R1 [DeepSeek-AI et al., 2025]. On the other hand, Yuan et al. [2023], Xie et al. [2025] utilizes rejection sampling fine-tuning. Specifically, the model will generate multiple reasoning traces, and the one that leads to the correct answer is selected for further fine-tuning. In this paper, we use human annotations as the SFT data (traditional SFT), as it is sufficient for our purpose and keeps the focus on our main contribution (unifying SFT and RFT).

**Reinforcement Fine-Tuning (RFT) for Reasoning.**   RFT for reasoning can be categorized into process supervision and outcome supervision. Process supervision assigns a reward to each step of a long reasoning trace [Lightman et al., 2024], which evaluates whether each step is correct or not. The main drawback of process supervision is that it is costly to prepare step-by-step feedback data. On the other hand, outcome supervision assigns a single reward to the entire trace [DeepSeek-AI et al., 2025, Zeng et al., 2025, Yu et al., 2025], *e.g.*, whether the trace yields the correct answer to a math problem. Furthermore, Wang et al. [2023], Yuan et al. [2024], Zhong et al. [2024], Luo et al. [2024], Setlur et al. [2025] learn a step-by-step reward model from a collection of reasoning traces with outcome rewards, which avoids the cost of preparing step-by-step data. In this paper, due to the efficiency and simplicity of outcome supervision, we focus on the comparison with RFT using outcome supervision.

**Curriculum Learning for Reasoning.**   Existing curriculum reinforcement learning for reasoning mainly focuses on utilizing a collection of problems with varying difficulties [Wen et al., 2025, Shi et al., 2025, Song et al., 2025]. These methods train the model with problems of gradually increasing difficulty, where the difficulty is determined by predefined criteria, such as the length of the successful reasoning trace [Song et al., 2025] or the success rate of baseline models [Shi et al., 2025, Wen et al., 2025]. However, such methods fail when the problems in the dataset are homogeneous in difficulty. In contrast, Xi et al. [2024] proposes a curriculum learning method that concatenates the problem with a slice of the solution (hint). The difficulty is determined by the hint length. However, Xi et al. [2024] uses a uniform distribution over all possible hint lengths, which misaligns with the distribution of interest (zero hint length). On the other hand, UFT designs a hint length scheduler that smoothly reduces the hint length to zero. Furthermore, UFT adds an additional log-likelihood term for the hint in the objective function, which helps the model to acquire new knowledge more efficiently and increases the ceiling of reinforcement learning (cf. Figure 5).

# B  Experiment Details

In this section, we describe our experimental setup and results. We present the pseudocode for UFT (Appendix B.1), detail the hyperparameters used (Appendix B.2), and conduct an ablation study (Appendix B.3). We also provide an analysis of the model's generalization to larger scales and report additional experimental findings (Appendix B.5).

## B.1  Algorithm

The pseudo-code of UFT is presented in Algorithm 1. In lines 4-9: we sample the hint length for each (question, solution, answer) pair in the sampled data batch $\mathcal{B}$. In lines 11-13, we concatenate the question with the partial solution of length $l(t)$ and feed it into a reinforcement learning algorithm (such as GRPO), with the objective function (3.3). Section 4 discusses the theoretical properties of UFT. To facilitate concrete convergence analysis, we specify the update rule of UFT, and the theoretically grounded variant is provided in Algorithm 2.

---
**Algorithm 1:** Unified Fine-Tuning
___
**Hyperparameters:** KL-penalty coefficient $\beta$, total number of steps $T$, number of steps with hint $T_{\text{hint}}$, low/high probability $p^{\text{low}}/p^{\text{high}}$ for hint sampling, and hint length $L$

**Input:** Reference policy parameter $\boldsymbol{\theta}^{\text{ref}}$

**Initialization:** $\boldsymbol{\theta}^{(0)} \leftarrow \boldsymbol{\theta}^{\text{ref}}$

1 **for** $t = 0, 1, \cdots, T - 1$ **do**
2    Sample a batch of problems $\mathcal{B}$
3    $\mathcal{D} \leftarrow \{\}$
4    **for** $(Q, S, A) \in \mathcal{B}$ **do**
      `// For each (question, solution, answer) pair`
5       **if** $t < T_{\text{hint}}$ **then**
6

$$p^{(t)} \leftarrow p^{\text{low}} + \frac{1}{2}\left(p^{\text{high}} - p^{\text{low}}\right)\left(1 + \cos\left(\frac{t+1}{T_{\text{hint}}}\pi\right)\right) \qquad \text{(B.1)}$$

        `// Cosine annealing, π ≈ 3.14159 is the Pi constant`
7         Sample $l^{(t)} \sim \text{Binomial}\left(\min\{L, \text{len}(S)\}, p^{(t)}\right)$
8       **else**
9         $l^{(t)} = 0$
10       **end**
11       $\mathcal{D} \leftarrow \mathcal{D} \cup \{Q + S[: l^{(t)}]\}$ `// Concatenate the question with the partial solution (hint) and add to` $\mathcal{D}$
12    **end**
13    Run reinforcement learning algorithm on $\mathcal{D}$ with the objective function (3.3)
14 **end**
___

## B.2 Cost and Implementation Details

The project costs roughly $10,000 GPU hours. The experiment is based on VERL [Sheng et al., 2024] and TinyZero [Pan et al., 2025]. The hyperparameters for training on different datasets are listed in Table 1. The omitted hyperparameters follow the default values of VERL [Sheng et al., 2024].

Additionally, we provide an illustration of a hint in MATH(3,4,5). Basically, we divide the whole solution by sentences, then uniformly divide the sentences into $L$ buckets. Then, during training, we will sample $l$ to decide the hint length (how many buckets included).

```
[
"For the piecewise function to be continuous, the cases must \"meet\" at $2$ and $-2$. ",
"For example, $ax+3$ and $x-5$ must be equal when $x=2$. ",
"This implies $a(2)+3=2-5$, which we solve to get $2a=-6 \\Rightarrow a=-3$. ",
"Similarly, $x-5$ and $2x-b$ must be equal when $x=-2$. ",
"Substituting, we get $-2-5=2(-2)-b$, which implies $b=3$. ",
"So $a+b=-3+3=\\boxed{0}$."
]
```

## B.3 Ablation Study

**Ablation on Hint Length.** We conducted a study on Qwen2.5-0.5B (MATH(3,4,5)), testing dividing the solution into $L = 4/5/6$ pieces uniformly and sampling hint length among $\{0, 1, \ldots, L\}$. We choose MATH(3,4,5) instead of Countdown because the solution length of MATH(3,4,5) is relatively longer. The results are shown in Table 2.

| Data | |
|---|---|
| Training Batch Size | 256 |
| Validation Batch Size | 1312 |
| Mini-batch Size | 64 |
| Hint Length | 5 |
| **Training** | |
| Learning Rate | $10^{-6}$ |
| $\beta$ | 0.001 |
| $T$ | 500 |
| $T_{\text{hint}}$ | 300 |
| Number of Rollouts | 4 |
| Context Window (Prompt) | Countdown: 256
MATH(3,4,5): 1024
Logic: 1024 |
| Context Window (Response) | 1024 |
| $p^{\text{low}}$ | 0.05 |
| $p^{\text{high}}$ | 0.95 |
| SFT Epochs | 5 |
| **Reward** | |
| Accuracy Reward | 1.0 |
| Format Correctness Reward | 0.1 |
| Incorrect Reward | 0.0 |

Table 1: The hyperparameters for training on different datasets. The other parameters follow the default parameters of VERL [Sheng et al., 2024].

| Hint Length Setting | Accuracy (%) |
|---|---|
| Hint length $L = 4$ | 27.44 |
| Hint length $L = 5$ | 27.15 |
| Hint length $L = 6$ | 25.20 |

Table 2: Ablation on hint length for Qwen2.5-0.5B (MATH(3,4,5)).

This suggests that UFT is relatively insensitive to the total number of pieces we divided the solution into.

**Ablation on $\beta$.** We evaluated different $\beta$ values on Qwen2.5-0.5B (Countdown). The results are shown in Table 3.

| $\beta$ | Accuracy (%) |
|---|---|
| $\beta = 0.0005$ | 46.09 |
| $\beta = 0.001$ | 50.39 |
| $\beta = 0.002$ | 59.95 |

Table 3: Ablation on $\beta$ for Qwen2.5-0.5B (Countdown).

A higher $\beta$ amplifies the impact of the supervised log-likelihood term, helping the model learn more from hints. For all main experiments, we adopt $\beta = 0.001$, the default in VERL. For larger models, our preliminary results show that varying $\beta$ yields minimal performance changes, likely due to their stronger pretrained priors. Table 4 presents the results of Qwen2.5-3B (Countdown).

| $\beta$ | Accuracy (%) |
|---|---|
| $\beta = 0.0005$ | 77.93 |
| $\beta = 0.001$ | 75.59 |
| $\beta = 0.002$ | 78.22 |

Table 4: Ablation on $\beta$ for Qwen2.5-3B (Countdown).

**Ablation on Hint Phase Length.** We test different durations for the hint phase on Qwen2.5-0.5B (Countdown), as shown in Table 5.

| Hint Phase Length | Accuracy (%) |
|---|---|
| $T_{\text{hint}} = 200$ | 52.15 |
| $T_{\text{hint}} = 300$ | 50.39 |
| $T_{\text{hint}} = 400$ | 50.00 |

Table 5: Ablation on hint phase length ($T_{\text{hint}}$) for Qwen2.5-0.5B (Countdown).

These small variations confirm that UFT is robust to the hint phase length.

**Ablation on Hint Length Distribution.** We also evaluate UFT under a two-point hint length sampling scheme: for any expected hint length $p \cdot L \in [n, n + 1)$, we set $l = n$ with probability $n + 1 - p \cdot L$ and $l = n + 1$ with probability $p \cdot L - n$. $p$ is still determined by the cosine-annealing scheduler. Further details are provided in Table 6.

| Hint Length Distribution | Accuracy (%) |
|---|---|
| Two-point | 48.63 |
| Binomial | 50.39 |

Table 6: Ablation on hint length distribution for Qwen2.5-0.5B (Countdown).

We can see that UFT is also robust to different choices of hint length distributions.

## B.4 Generalization to Larger Models

Even though we cannot afford training a larger model, we believe UFT will be helpful for the state-of-the-art models on tasks out of its capacity. For instance, some tasks involving knowledge of a special sub-field or extremely complex reasoning problems.

The following empirical results (Table 7) may hint at this. For instance, on the logic puzzle, which requires more abstract reasoning, Qwen2.5-1.5B performs poorly with RFT alone (0.00%) but achieves 23.00% with UFT:

| Task (Qwen2.5-1.5B) | RFT | UFT |
|---|---|---|
| Countdown | 71.48% | 71.48% |
| MATH(3,4,5) | 46.68% | 47.95% |
| Logic | 0.00% | 23.00% |

Table 7: Performance comparison between RFT and UFT across tasks with Qwen2.5-1.5B.

In conclusion, although Qwen2.5-1.5B has enough capacity for Countdown and MATH(3,4,5), it is insufficient in solving the logic puzzle and UFT outperforms RFT by a large margin. Therefore, it is likely that for larger models, UFT will be helpful when solving extremely hard tasks.

This stark difference in Logic suggests that even when a model appears to have "enough capacity" (as Qwen2.5-1.5B does for Countdown and MATH), it can still benefit significantly from UFT when the task challenges its reasoning ability. Hence, for much larger models facing extremely complex tasks, we anticipate that UFT will continue to provide meaningful advantages.

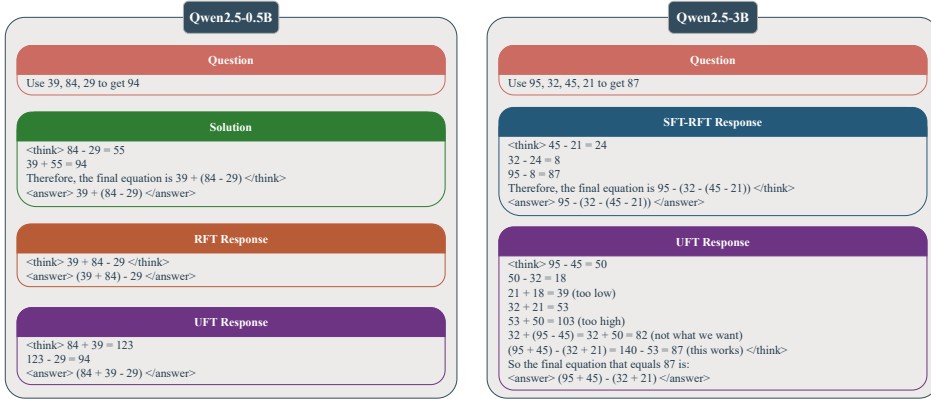

Figure 11: Responses of Qwen2.5-0.5/3B trained by different algorithms.

## B.5 Additional Results

Figure 11 shows the response of the model trained via different algorithms. For Qwen2.5-0.5B, UFT's response aligns with the solution better than RFT's. For Qwen2.5-3B, UFT generates a longer reasoning trace and presents skills such as verification [Gandhi et al., 2025], while SFT-RFT does not.

Table 8 shows the accuracy results across different datasets. For clarity, we report the average accuracy over models trained on three datasets: Countdown, MATH(3,4,5), and Logic.

For smaller models such as Qwen2.5-0.5B, SFT-RFT achieves an accuracy of 7.28%, compared to only 3.25% for RFT. In contrast, UFT achieves 9.45% accuracy, outperforming both.

For larger models such as Qwen2.5-3B, SFT-RFT achieves 17.34% accuracy, which is significantly lower than RFT's 32.15%. However, UFT still performs competitively, reaching 30.93% and closely matching RFT.

In summary, UFT combines the strengths of both SFT and RFT. When the model is small and memorization plays a key role, UFT matches or exceeds SFT's performance. When the model is large and generalization becomes more important, UFT benefits similarly to RFT, achieving comparable accuracy.

## C  Proof of Theorem 4.2

**Theorem 4.2** (Lowerbound). *For any integers $H \geq 1, B \geq 2$, and any RFT algorithm, there exists a problem with height $H$ and branching factor $B$, that satisfies the following: to achieve a $50\%$ pass@1 success rate, the algorithm needs to explore at least*

$$\frac{B^H}{4} \tag{4.3}$$

*nodes in $\mathcal{S}_H$. Moreover, when there are multiple nodes in $\mathcal{S}_H$ representing the correct solutions, e.g., $K \geq 1$, any algorithm needs to explore at least $\frac{B^H}{4K}$ nodes in $\mathcal{S}_H$.*

*Proof.* Proving the lower bound of exploration is equivalent to the following. Find the maximum $T > 0$, such that any algorithm will fail to learn the optimal policy with probability at least $0.5$ within $T$ explorations. Consider the $\binom{B^H}{K}$ possible trees, each associated with a distinct subset of $\mathcal{S}_H$ of size $K$, where that subset represents the correct solution for that specific tree. At the beginning, we pick an instance from all those possible trees uniformly at random.

During each exploration, the algorithm requests the reward at a node in $\mathcal{S}_H$. Let $s^{(1)}, s^{(2)}, \ldots, s^{(T)}$ be the leaf node reached at timestep $1, 2, \ldots T$, which are random variables depending on the randomness of the algorithm. Let $\mathcal{S}_H^* := \{s \in \mathcal{S}_H : \mathcal{R}(s) = \max_{s' \in \mathcal{S}_H} \mathcal{R}(s')\}$ be the set of nodes representing correct solutions. Note that given the construction of the instances, $|\mathcal{S}_H^*| = K$. Then, the probability

| Model | Algorithm | MATH(3,4,5) | AIME24 | AMC | Countdown | Logic | MATH500 | Minerva | Olympiad | GSM8k | Avg. |
|---|---|---|---|---|---|---|---|---|---|---|---|
| | Base | 3.03 | 0.00 | 0.00 | 0.00 | 0.00 | 1.73 | 0.74 | 0.30 | 7.66 | 1.55 |
| | SFT | 4.92 | 0.00 | 1.61 | 11.20 | 1.87 | 2.13 | 2.08 | 1.33 | 13.07 | 4.46 |
| | RFT | 3.78 | 0.00 | 3.21 | 8.30 | 0.00 | 2.47 | 3.80 | 2.57 | 3.87 | 3.25 |
| Qwen2.5-0.5B | SFT-RFT | 8.69 | 0.00 | 3.61 | **17.45** | **7.07** | 2.07 | 4.41 | 2.12 | 16.45 | 7.28 |
| | $R^3$ | 9.86 | 0.00 | 6.43 | 9.99 | 4.20 | 3.33 | 5.02 | **3.11** | 20.09 | 7.36 |
| | UFT | **13.18** | 0.00 | **6.83** | 17.15 | 4.87 | **5.40** | **5.76** | 2.77 | **24.59** | **9.45** |
| | Base | 24.51 | **3.33** | 4.82 | 0.20 | 2.20 | 18.27 | 4.41 | 5.48 | 60.96 | 14.29 |
| | SFT | 12.47 | 0.00 | 5.62 | 13.48 | 5.33 | 6.40 | 4.53 | 2.62 | 29.74 | 9.36 |
| | RFT | 24.77 | 2.22 | 9.24 | **27.86** | 3.00 | 10.53 | 6.86 | 6.47 | 45.69 | 16.08 |
| Qwen2.5-1.5B | SFT-RFT | 15.72 | 1.11 | 6.83 | 20.51 | 11.13 | 5.00 | 4.41 | 4.59 | 30.02 | 11.70 |
| | $R^3$ | 28.12 | 2.22 | 13.65 | 23.57 | **11.47** | 14.93 | 7.48 | 9.43 | 49.79 | 18.65 |
| | UFT | **34.08** | **3.33** | **14.86** | 24.54 | 10.07 | **20.87** | **8.33** | **9.68** | **66.46** | **22.23** |
| | Base | 31.45 | 0.00 | 13.25 | 3.81 | 5.60 | 24.53 | 4.78 | 7.70 | 57.85 | 17.13 |
| | SFT | 24.32 | 0.00 | 10.04 | 15.07 | 10.20 | 16.80 | 5.27 | 5.19 | 45.54 | 15.25 |
| | RFT | 45.74 | **4.44** | 24.90 | **34.08** | **30.33** | **31.27** | 12.25 | **15.65** | **80.84** | **32.15** |
| Qwen2.5-3B | SFT-RFT | 26.50 | 1.11 | 9.64 | 17.61 | 14.07 | 5.76 | 6.77 | | 48.22 | 17.34 |
| | $R^3$ | 44.01 | 2.22 | 21.29 | 27.12 | 24.80 | 28.00 | 10.91 | 14.57 | 70.20 | 28.02 |
| | UFT | **47.04** | 3.33 | **29.32** | 31.38 | 26.07 | 29.73 | **12.99** | 14.17 | 74.63 | 30.93 |
| | Base | 0.00 | 0.00 | 0.00 | 0.00 | 0.00 | 0.00 | 0.00 | 0.00 | 0.08 | 0.01 |
| | SFT | 1.07 | 0.00 | 0.80 | 13.41 | 3.67 | 0.00 | 0.74 | 0.25 | **1.87** | 2.49 |
| | RFT | 0.94 | 0.00 | **2.41** | 0.00 | 0.00 | **0.47** | 0.49 | 0.84 | 1.42 | 0.80 |
| Llama-3.2-1B | SFT-RFT | 0.42 | 0.00 | 0.00 | **18.68** | **8.33** | 0.00 | 1.23 | 0.20 | 0.48 | 3.29 |
| | $R^3$ | **1.53** | 0.00 | 1.61 | 9.90 | 0.13 | 0.33 | **2.94** | **0.99** | 1.49 | 2.20 |
| | UFT | 1.17 | 0.00 | 0.00 | 17.87 | 7.40 | 0.07 | 2.82 | 0.74 | 1.14 | **3.52** |
| | Base | 0.00 | 0.00 | 0.00 | 0.00 | 0.00 | 0.00 | 0.00 | 0.00 | 0.00 | 0.00 |
| | SFT | 2.54 | 0.00 | 0.40 | 14.68 | 6.13 | 0.00 | 1.72 | 0.54 | **7.08** | 3.85 |
| | RFT | 0.00 | 0.00 | 0.00 | 0.00 | 0.00 | 0.00 | 0.00 | 0.05 | 0.00 | 0.01 |
| Llama-3.2-3B | SFT-RFT | **3.16** | 0.00 | 2.41 | 16.05 | 8.87 | 0.07 | **3.92** | 0.89 | 5.76 | 4.79 |
| | $R^3$ | 2.93 | 0.00 | **3.21** | 17.55 | **9.93** | 0.87 | 3.06 | **1.04** | 5.16 | **5.03** |
| | UFT | 1.24 | 0.00 | 1.20 | **17.64** | 6.60 | **1.13** | 1.10 | 0.30 | 4.12 | 3.72 |

Table 8: Average performance of Qwen2.5-0.5/1.5/3B and Llama-3.2-1/3B across all three training datasets, Countdown, MATH(3,4,5), and Logic.

of reaching one of the correct solutions in $\mathcal{S}_H^*$ is

$$
\Pr\left(\left\{s^{(t)}\right\}_{t=1}^{T} \cap \mathcal{S}_H^* \neq \emptyset\right) = \sum_{t=1}^{T} \Pr\left(s^{(t)} \in \mathcal{S}_H^* \mid \left\{s^{(s)}\right\}_{s=1}^{t-1} \cap \mathcal{S}_H^* = \emptyset\right) \Pr\left(\left\{s^{(s)}\right\}_{s=1}^{t-1} \cap \mathcal{S}_H^* = \emptyset\right)
$$

$$
\leq \sum_{t=1}^{T} \Pr\left(s^{(t)} \in \mathcal{S}_H^* \mid \left\{s^{(s)}\right\}_{s=1}^{t-1} \cap \mathcal{S}_H^* = \emptyset\right).
$$

Given that we pick $\mathcal{S}_H^*$ uniformly at random, $\Pr\left(s^{(t)} \in \mathcal{S}_H^* \mid \left\{s^{(s)}\right\}_{s=1}^{t-1} \cap \mathcal{S}_H^* = \emptyset\right) = \frac{|\mathcal{S}_H^*|}{B^H - t + 1}$. Therefore,

$$
\Pr\left(\left\{s^{(t)}\right\}_{t=1}^{T} \cap \mathcal{S}_H^* \neq \emptyset\right) \leq \sum_{t=1}^{T} \frac{|\mathcal{S}_H^*|}{B^H - t + 1}.
$$

When $T \leq \frac{B^H}{4|\mathcal{S}_H^*|}$, we have

$$
\Pr\left(\left\{s^{(t)}\right\}_{t=1}^{T} \cap \mathcal{S}_H^* \neq \emptyset\right) \leq \sum_{t=1}^{T} \frac{|\mathcal{S}_H^*|}{B^H - t + 1} \overset{(i)}{\leq} \sum_{t=1}^{T} \frac{2|\mathcal{S}_H^*|}{B^H} = \frac{2|\mathcal{S}_H^*|T}{B^H} \leq \frac{1}{2}.
$$

$(i)$ uses the fact that $t \leq T \leq \frac{B^H}{4|\mathcal{S}_H^*|} \leq \frac{B^H}{2}$. Therefore, within $\frac{B^H}{4|\mathcal{S}_H^*|}$ exploration, the algorithm will fail to find the correct answer with probability at least $0.5$. □

# D Extended Theoretical Justifications

In this section, we introduce some additional notations in Appendix D.1 and then present the theoretically sound UFT in Appendix D.2.

### D.1 Extended Preliminaries

**Notation.** For any vector $\boldsymbol{x} \in \mathbb{R}^n$, let $x_i$ be its $i^{th}$ element and $\|\boldsymbol{x}\|_p$ be the $L_p$-norm, where $\|\boldsymbol{x}\|$ denotes the $L_2$-norm by default. For any two vectors $\boldsymbol{x}, \boldsymbol{y} \in \mathbb{R}^n$, let $\langle \boldsymbol{x}, \boldsymbol{y} \rangle := \sum_{i=1}^n x_i \cdot y_i$ denote their inner product.

**Softmax Parameterized Policy.** Algorithm 2 assumes the policy follows softmax parameterization. Formally, the policy $\pi^{\boldsymbol{\theta}}$ is controlled by $\boldsymbol{\theta} \in \mathbb{R}^{|\mathcal{S}| \times B}$, such that for any $s \in \mathcal{S}$ and $a \in [B]$,

$$\pi^{\boldsymbol{\theta}}(a \mid s) := \frac{\exp(\theta(s, a))}{\sum_{a'=1}^B \exp(\theta(s, a'))}. \tag{D.1}$$

The softmax-parameterized policy is also widely adopted in the literature [Mei et al., 2020, Agarwal et al., 2021, Ding et al., 2020] to sidestep the complexities of analyzing non-convex neural networks and to keep the focus on the learning algorithm itself.

### D.2 Theoretically Sound UFT

The full algorithm is shown in Algorithm 2. In lines 2-3: we sample the hint length and a trajectory starting from the hint. In lines 6-10, we estimate Q-values by sampling an additional trajectory for each state-action pair, which can greatly reduce the variance of sampling. In lines 13-14, we compute the objective function and update the parameters by gradient ascent. In lines 16-17, we estimate the expected reward of each intermediate policy and return the best one.

Note that Algorithm 2 differs slightly from the UFT shown in Algorithm 1. While Algorithm 1 leaves the choice of the reinforcement learning algorithm unspecified, Algorithm 2 explicitly defines the trajectory rolling mechanism and update rule for concrete theoretical analysis. Further, Algorithm 2 assumes a softmax-parameterized policy, whereas Algorithm 1 imposes no constraints on the policy network architecture.

## E Proof of Theorem 4.3

In this section, for notational simplicity, we use $\pi^{(t)}$ to denote $\pi^{\boldsymbol{\theta}^{(t)}}$ for any $t \in \{0, 1, \cdots, T\}$. Moreover, for any $t \in [T]$, we define $\widetilde{A}^{(t-1)}(s, a) = \widetilde{Q}^{(t-1)}(s, a) = 0$ for those nodes $s$ off the sampled path $\left( s_h^{(t)} \right)_{h=l^{(t)}}^H$ at timestep $t$.

**Theorem E.1** (Formal). Consider Algorithm 2. When $\beta \leq \frac{\Delta}{12(H+1)^2 \left( \log B + 2 \|\boldsymbol{\theta}^{\text{ref}}\|_\infty \right)}$, the pass @ 1 accuracy $\Pr_{\pi^{\boldsymbol{\theta}^{(\tilde{t}^*)}}}$ (pass @ 1) of policy $\pi^{\boldsymbol{\theta}^{(\tilde{t}^*)}}$ satisfies

$$\Pr_{\pi^{\boldsymbol{\theta}^{(\tilde{t}^*)}}} (\text{pass @ 1}) \geq 0.5, \tag{E.1}$$

when

$$T = \left( \frac{(H+1)^2 \left( \log B + 2 \|\boldsymbol{\theta}^{\text{ref}}\|_\infty + 7 \right)}{\Delta/12} \right)^2 \tag{E.2}$$

and explores no more than $(BH + N)T$ leaf nodes in $\mathcal{S}_H$.

*Proof.* The update rule can be divided into two steps: (i) Use the concentration bound to get a high-probability bound on $\left\langle Q^{\pi^{(t-1)}}(s, \cdot), \pi^*(\cdot \mid s) - \pi^{(t-1)}(\cdot \mid s) \right\rangle$ (cf. Appendix E.1); (ii) Convert the difference in each node to the $V^* - V^{\pi^{(t-1)}}(s_{\text{root}})$ by the regret decomposition lemma (cf. Appendix E.2); (iii) Convert the bound on expected reward to success rate (cf. Appendix E.3).

### E.1 Concentration Bound

For any height $h \in \{0\} \cup [H-1]$, state $s \in \mathcal{S}_h$, and action $a \in [B]$, we can define the Q-value of the state-action pair $(s, a) \in \mathcal{S} \times [B]$ when following policy $\pi$ as

$$Q^\pi(s, a) := \mathbb{E}_{s_h = s, (s_{h'})_{h'=h}^H \sim \pi} \left[ \mathcal{R} \left( s_H \right) \right]. \tag{E.3}$$

---

**Algorithm 2:** Theoretically Sound Unified Fine-Tuning

---

**Hyperparameters:** Learning rate $\eta$, KL-penalty coefficient $\beta$, and total number of steps $T$

**Input:** Reference policy parameter $\boldsymbol{\theta}^{\text{ref}}$

**Initialization:** $\boldsymbol{\theta}^{(0)} \leftarrow \boldsymbol{\theta}^{\text{ref}}$

1 **for** $t = 0, 1, \cdots, T-1$ **do**

2     Sample $l^{(t)} \sim \text{Uniform}(0, 1, 2, \cdots, H-1, H)$

     `// In fact, any distribution with full support on {0,1,2,···,H-1,H} is`
        `fine. We choose the uniform distribution for simplicity`

3     Sample trajectory $\left(s_h^{(t)}\right)_{h=l^{(t)}}^{H} \sim \pi^{\boldsymbol{\theta}^{(t)}}$, where $s_{l^{(t)}}^{(t)} = s_{l^{(t)}}^{*}$

4     **for** $h = l^{(t)}, l^{(t)}+1, \cdots H-1$ **do**

5        **for** $a = 1, 2, \cdots, B$ **do**

          `// Group sampling`

6           Sample trajectory $\left(s_{h'}^{(t),a}\right)_{h'=h+1}^{H} \sim \pi^{\boldsymbol{\theta}^{(t)}}$ starting from $s_{h+1}^{(t),a} = \mathcal{T}(s_h^{(t)}, a)$

7           $\widetilde{Q}^{(t)}\left(s_h^{(t)}, a\right) \leftarrow \mathcal{R}\left(s_H^{(t),a}\right)$

8        **end**

9        **for** $a = 1, 2, \cdots, B$ **do**

10         $\widetilde{A}^{(t)}\left(s_h^{(t)}, a\right) \leftarrow \widetilde{Q}^{(t)}\left(s_h^{(t)}, a\right) - \sum_{a=1}^{B} \pi^{\boldsymbol{\theta}^{(t)}}\left(a \mid s_h^{(t)}\right) \widetilde{Q}^{(t)}\left(s_h^{(t)}, a\right)$

11        **end**

12     **end**

     `// ` $\widetilde{A}^{(t)}(s, \cdot) \equiv 0$ ` for any ` $s$ ` off the trajectory ` $\left(s_h^{(t)}\right)_{h=l^{(t)}}^{H}$

13

$$
\mathcal{J}^{(t)} \leftarrow \sum_{h=l^{(t)}}^{H-1} \sum_{a=1}^{B} \pi^{\boldsymbol{\theta}^{(t)}}\left(a \mid s_h^{(t)}\right) \widetilde{A}^{(t)}\left(s_h^{(t)}, a\right)
$$
$$
- \beta \sum_{h=l^{(t)}}^{H-1} \text{KL}\left(\pi^{\boldsymbol{\theta}^{(t)}}\left(\cdot \mid s_h^{(t)}\right) \| \pi^{\boldsymbol{\theta}^{\text{ref}}}\left(\cdot \mid s_h^{(t)}\right)\right) + \beta \sum_{h=0}^{l^{(t)}-1} \log \pi^{\boldsymbol{\theta}^{(t)}}\left(a_h^{*} \mid s_h^{*}\right)
$$

14

$$
\boldsymbol{\theta}^{(t+1)} \leftarrow \boldsymbol{\theta}^{(t)} + \eta \nabla_{\pi} \mathcal{J}^{(t)} \tag{D.2}
$$

15 **end**

16 Estimate $\widetilde{V}^{\pi^{\boldsymbol{\theta}^{(t)}}}(s_{\text{root}}) = \frac{1}{N} \sum_{n=1}^{N} \mathcal{R}\left(\widetilde{s}_H^{(t),n}\right)$ by sampling trajectories $\widetilde{s}_0^{(t),n} = s_{\text{root}}$ and

    $\left(\widetilde{s}_h^{(t),n}\right)_{h=0}^{H} \sim \pi^{\boldsymbol{\theta}^{(t)}}$, where $N = \frac{72 \log(14(T+1))}{\Delta^2}$

17 $\widetilde{t}^{*} = \text{argmax}_{t \in \{0,1,\cdots,T\}} \widetilde{V}^{\pi^{\boldsymbol{\theta}^{(t)}}}(s_{\text{root}})$

**Return:** $\pi^{\boldsymbol{\theta}^{(\widetilde{t}^{*})}}$

---

Then, for any $s \in \mathcal{S} \setminus \mathcal{S}_H$ and $t \in [T]$, we have

$$
\mathbb{E}\left[\widetilde{Q}^{(t-1)}(s, a)\right] = \text{Pr}\left(s \in \left\{s_h^{(t)}\right\}_{h=l^{(t)}}^{H}\right) \cdot Q^{\pi^{(t-1)}}(s, a), \tag{E.4}
$$

where the expectation is taken over the probability of sampling trajectories in Algorithm 2. Next, we will introduce Lemma 5.3 in Liu et al. [2024].

**Proposition E.2.** Let $M, \widetilde{M} \geq 0$ be the constants such that $\left|f^{(t)}(\boldsymbol{x}) - f^{(t)}(\boldsymbol{x}')\right| \leq M$ and $\left|\widetilde{f}^{(t)}(\boldsymbol{x}) - \widetilde{f}^{(t)}(\boldsymbol{x}')\right| \leq \widetilde{M}$ for any $t \in [T]$ and $\boldsymbol{x}, \boldsymbol{x}' \in \mathcal{C}$, where $\mathcal{C}$ is a convex set. If for any

$\boldsymbol{x} \in \mathcal{C}$, we have

$$\mathbb{E}\left[\widetilde{f}^{(t)}(\boldsymbol{x}) \mid \widetilde{f}^{(1)}, \widetilde{f}^{(2)}, \cdots, \widetilde{f}^{(t-1)}\right] = f^{(t)}(\boldsymbol{x}),$$

and $\boldsymbol{x}^{(t)}$ is deterministically influenced by $\widetilde{f}^{(1)}, \widetilde{f}^{(2)}, \cdots, \widetilde{f}^{(t-1)}$, then for any $\delta \in (0, 1)$ and $\boldsymbol{x} \in \mathcal{C}$, we have

$$\Pr\left(\sum_{t=1}^{T}\left(f^{(t)}(\boldsymbol{x}) - f^{(t)}(\boldsymbol{x}^{(t)})\right) \le \sum_{t=1}^{T}\left(\widetilde{f}^{(t)}(\boldsymbol{x}) - \widetilde{f}^{(t)}(\boldsymbol{x}^{(t)})\right) + \left(M + \widetilde{M}\right)\sqrt{2T\log\frac{1}{\delta}}\right) \ge 1 - \delta.$$

For any $h < H$ and $s \in \mathcal{S}_h$, let $f^{(t)}(\boldsymbol{x}) = \Pr\left(s \in \left\{s_h^{(t-1)}\right\}_{h=l^{(t-1)}}^{H}\right)\left\langle Q^{\pi^{(t-1)}}(s, \cdot), \boldsymbol{x}\right\rangle$, where $f^{(t)}: \Delta^B \to [0, 1]$ since each element of $Q^{(t-1)}(s, \cdot)$ is bounded by $[0, 1]$ by definition. Therefore, $M$ in Proposition E.2 is 1. Similarly, let $\widetilde{f}^{(t)}(\boldsymbol{x}) = \left\langle \widetilde{Q}^{(t-1)}(s, \cdot), \boldsymbol{x}\right\rangle$ and we have $\widetilde{M} = 1$. Therefore, by (E.4), Proposition E.2, and Lemma E.3, for any $\delta \in (0, 1)$, with probability at least $1 - \delta$, we have

$$\sum_{t=1}^{T}\Pr\left(s \in \left\{s_h^{(t)}\right\}_{h=l^{(t)}}^{H}\right)\left\langle Q^{\pi^{(t-1)}}(s, \cdot), \pi^*(\cdot \mid s) - \pi^{(t-1)}(\cdot \mid s)\right\rangle$$

$$\le \sum_{t=1}^{T}\left\langle \widetilde{Q}^{(t-1)}(s, \cdot), \pi^*(\cdot \mid s) - \pi^{(t-1)}(\cdot \mid s)\right\rangle + 2\sqrt{2T\log\frac{1}{\delta}}$$

$$\overset{(i)}{=} \sum_{t=1}^{T}\left\langle \widetilde{A}^{(t-1)}(s, \cdot), \pi^*(\cdot \mid s) - \pi^{(t-1)}(\cdot \mid s)\right\rangle + 2\sqrt{2T\log\frac{1}{\delta}}.$$

$(i)$ is because

$$\left\langle \widetilde{A}^{(t-1)}(s, \cdot), \pi^*(\cdot \mid s) - \pi^{(t-1)}(\cdot \mid s)\right\rangle$$

$$= \left\langle \widetilde{Q}^{(t-1)}(s, \cdot), \pi^*(\cdot \mid s) - \pi^{(t-1)}(\cdot \mid s)\right\rangle$$

$$+ \sum_{a=1}^{B}\pi^{(t-1)}(a \mid s)\widetilde{Q}^{(t-1)}(s, a)\sum_{a=1}^{B}\left(\pi^*(a \mid s) - \pi^{(t-1)}(a \mid s)\right)$$

$$= \left\langle \widetilde{Q}^{(t-1)}(s, \cdot), \pi^*(\cdot \mid s) - \pi^{(t-1)}(\cdot \mid s)\right\rangle.$$

By the update rule of Algorithm 2, we have the following lemma.

**Lemma E.3.** Consider Algorithm 2. For any node $s \in \mathcal{S} \setminus \mathcal{S}_H$, we have

$$\sum_{t=1}^{T}\left\langle \widetilde{A}^{(t-1)}(s, \cdot), \pi^*(\cdot \mid s) - \pi^{(t-1)}(\cdot \mid s)\right\rangle$$

$$\le \left(\frac{1}{\eta} + \beta T\right)\mathrm{KL}\left(\pi^*(\cdot \mid s)\|\pi^{\boldsymbol{\theta}^{\mathrm{ref}}}(\cdot \mid s)\right) + 2\eta T.$$

The proof is postponed to Appendix E.4. Lemma E.3 gives us an upper bound on the accumulated difference between our policy $\pi^{(t-1)}$ and the optimal policy $\pi^*$. Therefore,

$$\sum_{t=1}^{T}\Pr\left(s \in \left\{s_h^{(t)}\right\}_{h=l^{(t)}}^{H}\right)\left\langle Q^{\pi^{(t-1)}}(s, \cdot), \pi^*(\cdot \mid s) - \pi^{(t-1)}(\cdot \mid s)\right\rangle$$

$$\le \sum_{t=1}^{T}\left\langle \widetilde{A}^{(t-1)}(s, \cdot), \pi^*(\cdot \mid s) - \pi^{(t-1)}(\cdot \mid s)\right\rangle + 2\sqrt{2T\log\frac{1}{\delta}}$$

$$\le \left(\frac{1}{\eta} + \beta T\right)\mathrm{KL}\left(\pi^*(\cdot \mid s)\|\pi^{\boldsymbol{\theta}^{\mathrm{ref}}}(\cdot \mid s)\right) + 2\eta T + 2\sqrt{2T\log\frac{1}{\delta}}.$$

## E.2 Difference Decomposition

Let $\mu^{\pi}(s)$ be the probability of reaching state $s$ from the root by following policy $\pi$. Hence, $\mu^{\pi}(s_{\text{root}}) = 1$. For any $s \in \mathcal{S} \setminus \mathcal{S}_H$ and action $a \in [B]$, $\mu^{\pi}(\mathcal{T}(s,a))$ can be recursively defined as

$$\mu^{\pi}(\mathcal{T}(s,a)) = \mu^{\pi}(s) \cdot \pi(s,a). \tag{E.5}$$

In the following, we will introduce Lemma E.4, which is a special case of the regret decomposition lemma (Lemma 5.1) in Liu et al. [2023]. Specifically, it is the regret decomposition lemma for a two-player zero-sum extensive-form game without chance nodes[5], and the second player's action sets at all nodes are of size 1.

**Lemma E.4.** For any sequence of policies $\pi^{(1)}, \pi^{(2)}, \cdots, \pi^{(T)}$ and policy $\pi$, we have

$$\sum_{t=1}^{T} \left( V^{\pi}(s_{\text{root}}) - V^{\pi^{(t)}}(s_{\text{root}}) \right) = \sum_{s \in \mathcal{S} \setminus \mathcal{S}_H} \mu^{\pi}(s) \sum_{t=1}^{T} \left\langle Q^{\pi^{(t)}}(s,\cdot), \pi(\cdot \mid s) - \pi^{(t)}(\cdot \mid s) \right\rangle.$$

Lemma E.4 can also be viewed as the performance difference lemma in reinforcement learning [Kakade and Langford, 2002] for a tree-shape Markov decision process. For completeness, we also provide the proof at the end of this section.

By letting $\pi^{(t)} = \pi^{(t-1)}$ for any $t \in [T]$ and $\pi = \pi^*$, we have

$$\sum_{t=1}^{T} \left( V^* - V^{\pi^{(t-1)}}(s_{\text{root}}) \right)$$

$$= \sum_{s \in \mathcal{S} \setminus \mathcal{S}_H} \mu^{\pi^*}(s) \sum_{t=1}^{T} \left\langle Q^{\pi^{(t-1)}}(s,\cdot), \pi^*(\cdot \mid s) - \pi^{(t-1)}(\cdot \mid s) \right\rangle$$

$$\overset{(i)}{=} \sum_{s \in \{s_0^*, s_1^*, \cdots, s_{H-1}^*\}} \mu^{\pi^*}(s) \sum_{t=1}^{T} \left\langle Q^{\pi^{(t-1)}}(s,\cdot), \pi^*(\cdot \mid s) - \pi^{(t-1)}(\cdot \mid s) \right\rangle$$

$$= \sum_{s \in \{s_0^*, s_1^*, \cdots, s_{H-1}^*\}} \sum_{t=1}^{T} \frac{\mu^{\pi^*}(s)}{\Pr\left( s \in \left\{ s_h^{(t)} \right\}_{h=l^{(t)}}^{H} \right)} \Pr\left( s \in \left\{ s_h^{(t)} \right\}_{h=l^{(t)}}^{H} \right)$$

$$\cdot \left\langle Q^{\pi^{(t-1)}}(s,\cdot), \pi^*(\cdot \mid s) - \pi^{(t-1)}(\cdot \mid s) \right\rangle.$$

$(i)$ uses the fact that $\pi^*$ is deterministic such that $\mu^{\pi^*}(s) > 0$ only when $s \in \{s_0^*, s_1^*, \cdots, s_H^*\}$. Since $s_{l^{(t)}}^{(t)}$ is sampled from $\{s_0^*, s_1^*, \cdots, s_H^*\}$ uniformly, for any $s \in \{s_0^*, s_1^*, \cdots, s_H^*\}$, we have

$$\Pr\left( s \in \left\{ s_h^{(t)} \right\}_{h=l^{(t)}}^{H} \right) \geq \Pr\left( s = s_{l^{(t)}}^{(t)} \right) = \frac{1}{H+1}.$$

Therefore, $\dfrac{\mu^{\pi^*}(s)}{\Pr\left( s \in \left\{ s_h^{(t)} \right\}_{h=l^{(t)}}^{H} \right)} \leq H+1$ and we have

$$\sum_{t=1}^{T} \left( V^* - V^{\pi^{(t-1)}}(s_{\text{root}}) \right)$$

$$\leq \sum_{h=0}^{H-1} \frac{\mu^{\pi^*}(s_h^*)}{\Pr\left( s_h^* \in \left\{ s_h^{(t)} \right\}_{h=l^{(t)}}^{H} \right)} \left( \left( \frac{1}{\eta} + \beta T \right) \mathrm{KL}\left( \pi^*(\cdot \mid s_h^*) \| \pi^{\boldsymbol{\theta}^{\mathrm{ref}}}(\cdot \mid s_h^*) \right) + 2\eta T + 2\sqrt{2T \log \frac{1}{\delta}} \right)$$

$$\leq (H+1) \sum_{h=0}^{H-1} \left( \left( \frac{1}{\eta} + \beta T \right) \mathrm{KL}\left( \pi^*(\cdot \mid s_h^*) \| \pi^{\boldsymbol{\theta}^{\mathrm{ref}}}(\cdot \mid s_h^*) \right) + 2\eta T + 2\sqrt{2T \log \frac{1}{\delta}} \right).$$

Next, we can bound $\mathrm{KL}\left( \pi^*(\cdot \mid s_h^*) \| \pi^{\boldsymbol{\theta}^{\mathrm{ref}}}(\cdot \mid s_h^*) \right)$ by the following lemma.

---

[5]Chance nodes represent the randomness of the game, such as rolling a dice.

**Lemma E.5.** For any $h \in \{0, 1, \cdots, H-1\}$, we have

$$\mathrm{KL}\left(\pi^*(\cdot \mid s_h^*) \| \pi^{\boldsymbol{\theta}^{\mathrm{ref}}}(\cdot \mid s_h^*)\right) \leq \log B + 2\left\|\boldsymbol{\theta}^{\mathrm{ref}}\right\|_\infty.$$

The proof is postponed to Appendix E.4.

Therefore, by taking $\eta = \frac{1}{\sqrt{T}}$, we have

$$\sum_{t=1}^{T}\left(V^* - V^{\pi^{(t-1)}}(s_{\mathrm{root}})\right)$$

$$\leq (H+1)^2 \left( \left(\log B + 2\left\|\boldsymbol{\theta}^{\mathrm{ref}}\right\|_\infty\right) \sqrt{T} + 2\sqrt{T} + 2\sqrt{2T \log \frac{1}{\delta}} \right)$$

$$+ \beta T (H+1) \sum_{h=0}^{H-1} \mathrm{KL}\left(\pi^*(\cdot \mid s_h^*) \| \pi^{\boldsymbol{\theta}^{\mathrm{ref}}}(\cdot \mid s_h^*)\right).$$

Because $V^* - V^{\pi^{(t-1)}}(s_{\mathrm{root}}) \geq 0$ for any $t \in [T]$, according to pigeon hole principle, there must exist $t^* \in \{0, 1, \ldots, T\}$ such that

$$V^* - V^{\pi^{(t^*)}}(s_{\mathrm{root}})$$

$$\leq \frac{(H+1)^2 \left( \left(\log B + 2\left\|\boldsymbol{\theta}^{\mathrm{ref}}\right\|_\infty\right) + 2 + 2\sqrt{2\log\frac{1}{\delta}} \right)}{\sqrt{T}}$$

$$+ \beta(H+1) \sum_{h=0}^{H-1} \mathrm{KL}\left(\pi^*(\cdot \mid s_h^*) \| \pi^{\boldsymbol{\theta}^{\mathrm{ref}}}(\cdot \mid s_h^*)\right).$$

For any $\epsilon > \beta(H+1)\sum_{h=0}^{H-1} \mathrm{KL}\left(\pi^*(\cdot \mid s_h^*) \| \pi^{\boldsymbol{\theta}^{\mathrm{ref}}}(\cdot \mid s_h^*)\right)$, it takes

$$\left( \frac{(H+1)^2 \left(\log B + 2\left\|\boldsymbol{\theta}^{\mathrm{ref}}\right\|_\infty + 2 + 2\sqrt{2\log\frac{1}{\delta}}\right)}{\epsilon - \beta(H+1)\sum_{h=0}^{H-1} \mathrm{KL}\left(\pi^*(\cdot \mid s_h^*) \| \pi^{\boldsymbol{\theta}^{\mathrm{ref}}}(\cdot \mid s_h^*)\right)} \right)^2$$

iterations to satisfy $V^* - V^{\pi^{(t^*)}}(s_{\mathrm{root}}) \leq \epsilon$.

Recall that $\Delta > 0$ is the sub-optimality gap. By picking $\epsilon = \frac{\Delta}{6}$, $\delta = \frac{1}{8}$, and $\beta \leq \frac{\Delta}{12(H+1)^2\left(\log B + 2\|\boldsymbol{\theta}^{\mathrm{ref}}\|_\infty\right)}$, to get $\epsilon$ accuracy with probability $1 - \delta$, we need

$$T = \left( \frac{(H+1)^2 \left(\log B + 2\left\|\boldsymbol{\theta}^{\mathrm{ref}}\right\|_\infty + 7\right)}{\Delta/12} \right)^2$$

iterations, which implies $T \leq \mathcal{O}\left(\frac{H^4(\log B)^2}{\Delta^2}\right)$. Since $\mathcal{O}(B \cdot H)$ leaf nodes are explored at each iteration, the number of leaf nodes explored during training is $\mathcal{O}(B \cdot H \cdot T) \leq \mathcal{O}\left(B\frac{H^5(\log B)^2}{\Delta^2}\right)$.

### E.3 Compute Probability

To find $t^*$, we need to estimate $V^{\pi^{\boldsymbol{\theta}^{(t)}}}$ for all $t \in \{0, 1, \cdots, T\}$ by sampling trajectories. By sampling a trajectory from $\pi^{\boldsymbol{\theta}^{(t)}}$, we can get a random variable from Bernoulli $\left(\mathrm{Pr}_{\pi^{(t)}}^{\mathrm{cond}}(\text{pass @ 1})\right)$ representing whether the trajectory reaches the correct solution. Then, by Hoeffding's inequality, by sampling $N$ trajectories, we have

$$\mathrm{Pr}\left( \left| \widetilde{V}^{\pi^{\boldsymbol{\theta}^{(t)}}}(s_{\mathrm{root}}) - V^{\pi^{\boldsymbol{\theta}^{(t)}}}(s_{\mathrm{root}}) \right| \leq \frac{\Delta}{12} \right) \leq 2\exp\left(-\frac{N\Delta^2}{72}\right) \overset{(i)}{=} \frac{1}{7(T+1)}. \tag{E.6}$$

$(i)$ is by definition of $N$ in Algorithm 2. By union bound, for any $t \in \{0, 1, \cdots, T\}$, $\left| \widetilde{V}^{\pi^{\boldsymbol{\theta}^{(t)}}}(s_{\text{root}}) - V^{\pi^{\boldsymbol{\theta}^{(t)}}}(s_{\text{root}}) \right| \leq \frac{\Delta}{12}$ holds with probability at least $1 - \frac{T+1}{7(T+1)} = \frac{6}{7}$. Therefore,

$$V^{\pi^{\boldsymbol{\theta}^{(\widetilde{t}^*)}}}(s_{\text{root}}) \geq \widetilde{V}^{\pi^{\boldsymbol{\theta}^{(\widetilde{t}^*)}}}(s_{\text{root}}) - \frac{\Delta}{12} \geq \widetilde{V}^{\pi^{\boldsymbol{\theta}^{(t^*)}}}(s_{\text{root}}) - \frac{\Delta}{12}$$

$$\geq V^{\pi^{\boldsymbol{\theta}^{(t^*)}}}(s_{\text{root}}) - \frac{\Delta}{6} \geq V^* - \epsilon - \frac{\Delta}{6} = V^* - \frac{\Delta}{3}.$$

Recall that $\Pr_{\pi^{(\widetilde{t}^*)}}(\text{pass @ 1})$ is the pass @ 1 accuracy of policy $\pi^{(\widetilde{t}^*)}$. In the following, we will use $\Pr^{\text{cond}}$ as a shorthand of $\Pr\left( \cdot \mid V^{\pi^{(\widetilde{t}^*)}}(s_{\text{root}}) \geq V^* - \frac{\Delta}{3} \right)$.

$$\Pr^{\text{cond}}_{\pi^{(\widetilde{t}^*)}}(\text{pass @ 1}) = \Pr^{\text{cond}}_{s_0 = s_{\text{root}}, (s_h)_{h=0}^H \sim \pi^{(\widetilde{t}^*)}} \left( \mathcal{R}(s_H) = \max_{s_H' \in \mathcal{S}_H} \mathcal{R}(s_H') \right)$$

$$= \Pr^{\text{cond}}_{s_0 = s_{\text{root}}, (s_h)_{h=0}^H \sim \pi^{(\widetilde{t}^*)}} \left( \mathcal{R}(s_H) = V^* \right).$$

Furthermore,

$$V^* - \frac{\Delta}{3} \leq V^{\pi^{(\widetilde{t}^*)}}(s_{\text{root}}) = \mathbb{E}_{s_0 = s_{\text{root}}, (s_h)_{h=0}^H \sim \pi^{(\widetilde{t}^*)}} [\mathcal{R}(s_H)]$$

$$\leq \Pr^{\text{cond}}_{s_0 = s_{\text{root}}, (s_h)_{h=0}^H \sim \pi^{(\widetilde{t}^*)}} \left( \mathcal{R}(s_H) = V^* \right) V^*$$

$$+ \left( 1 - \Pr^{\text{cond}}_{s_0 = s_{\text{root}}, (s_h)_{h=0}^H \sim \pi^{(\widetilde{t}^*)}} \left( \mathcal{R}(s_H) = V^* \right) \right) (V^* - \Delta).$$

By combining all pieces together, we have

$$\Pr^{\text{cond}}_{\pi^{(\widetilde{t}^*)}}(\text{pass @ 1}) V^* + \left( 1 - \Pr^{\text{cond}}_{\pi^{(\widetilde{t}^*)}}(\text{pass @ 1}) \right) (V^* - \Delta)$$

$$\geq V^* - \frac{\Delta}{3},$$

which implies that $\Pr^{\text{cond}}_{\pi^{(\widetilde{t}^*)}}(\text{pass @ 1}) \geq \frac{2}{3}$.

Finally,

$$\Pr_{\pi^{(\widetilde{t}^*)}}(\text{pass @ 1})$$

$$\geq \Pr^{\text{cond}}_{\pi^{(\widetilde{t}^*)}}(\text{pass @ 1}) \Pr\left( V^{\pi^{(t^*)}}(s_{\text{root}}) \geq V^* - \epsilon \right) \Pr\left( V^{\pi^{(\widetilde{t}^*)}}(s_{\text{root}}) \geq V^{\pi^{(t^*)}}(s_{\text{root}}) - \frac{\Delta}{6} \right)$$

$$\geq \frac{2}{3}(1 - \delta)\frac{6}{7} = \frac{1}{2}. \qquad \square$$

### E.4   Omitted Proofs

**Lemma E.3.** Consider Algorithm 2. For any node $s \in \mathcal{S} \setminus \mathcal{S}_H$, we have

$$\sum_{t=1}^T \left\langle \widetilde{A}^{(t-1)}(s, \cdot), \pi^*(\cdot \mid s) - \pi^{(t-1)}(\cdot \mid s) \right\rangle$$

$$\leq \left( \frac{1}{\eta} + \beta T \right) \text{KL}\left( \pi^*(\cdot \mid s) \| \pi^{\boldsymbol{\theta}^{\text{ref}}}(\cdot \mid s) \right) + 2\eta T.$$

*Proof.* We will introduce the following one-step analysis of the update rule first.

**Lemma E.6.** For any node $s \in \mathcal{S} \setminus \mathcal{S}_H$ and $t \in [T]$, we have

$$\eta \left\langle \widetilde{A}^{(t-1)}(s, \cdot), \pi^*(\cdot \mid s) - \pi^{(t)}(\cdot \mid s) \right\rangle$$

$$\leq \text{KL}\left( \pi^*(\cdot \mid s) \| \pi^{(t-1)}(\cdot \mid s) \right) - \text{KL}\left( \pi^*(\cdot \mid s) \| \pi^{(t)}(\cdot \mid s) \right) - \text{KL}\left( \pi^{(t)}(\cdot \mid s) \| \pi^{(t-1)}(\cdot \mid s) \right)$$

$$+ \eta\beta \text{KL}\left( \pi^*(\cdot \mid s) \| \pi^{\boldsymbol{\theta}^{\text{ref}}}(\cdot \mid s) \right).$$

The proof is presented later in this section. Therefore,

$$\eta \left\langle \widetilde{A}^{(t-1)}(s, \cdot), \pi^*(\cdot \mid s) - \pi^{(t)}(\cdot \mid s) \right\rangle$$

$$\leq \mathrm{KL}\left(\pi^*(\cdot \mid s) \| \pi^{(t-1)}(\cdot \mid s)\right) - \mathrm{KL}\left(\pi^*(\cdot \mid s) \| \pi^{(t)}(\cdot \mid s)\right) - \mathrm{KL}\left(\pi^{(t)}(\cdot \mid s) \| \pi^{(t-1)}(\cdot \mid s)\right)$$

$$+ \eta\beta\mathrm{KL}\left(\pi^*(\cdot \mid s) \| \pi^{\boldsymbol{\theta}^{\mathrm{ref}}}(\cdot \mid s)\right).$$

By adding $\eta \left\langle \widetilde{A}^{(t-1)}(s, \cdot), \pi^{(t)}(\cdot \mid s) - \pi^{(t-1)}(\cdot \mid s) \right\rangle$ on both sides, we have

$$\eta \left\langle \widetilde{A}^{(t-1)}(s, \cdot), \pi^*(\cdot \mid s) - \pi^{(t-1)}(\cdot \mid s) \right\rangle$$

$$\overset{(i)}{\leq} \mathrm{KL}\left(\pi^*(\cdot \mid s) \| \pi^{(t-1)}(\cdot \mid s)\right) - \mathrm{KL}\left(\pi^*(\cdot \mid s) \| \pi^{(t)}(\cdot \mid s)\right) - \mathrm{KL}\left(\pi^{(t)}(\cdot \mid s) \| \pi^{(t-1)}(\cdot \mid s)\right)$$

$$+ \eta \left\langle \widetilde{A}^{(t-1)}(s, \cdot), \pi^{(t)}(\cdot \mid s) - \pi^{(t-1)}(\cdot \mid s) \right\rangle + \eta\beta\mathrm{KL}\left(\pi^*(\cdot \mid s) \| \pi^{\boldsymbol{\theta}^{\mathrm{ref}}}(\cdot \mid s)\right).$$

By Hölder's inequality, we have

$$\left\langle \widetilde{A}^{(t-1)}(s, \cdot), \pi^{(t)}(\cdot \mid s) - \pi^{(t-1)}(\cdot \mid s) \right\rangle$$

$$\leq \left\| \widetilde{A}^{(t-1)}(s, \cdot) \right\|_\infty \cdot \left\| \pi^{(t)}(\cdot \mid s) - \pi^{(t-1)}(\cdot \mid s) \right\|_1$$

$$\leq 2\eta \left\| \widetilde{A}^{(t-1)}(s, \cdot) \right\|_\infty^2 + \frac{1}{8\eta} \left\| \pi^{(t)}(\cdot \mid s) - \pi^{(t-1)}(\cdot \mid s) \right\|_1^2$$

$$\overset{(i)}{\leq} 2\eta + \frac{1}{4\eta}\mathrm{KL}\left(\pi^{(t)}(\cdot \mid s) \| \pi^{(t-1)}(\cdot \mid s)\right).$$

$(i)$ uses $\left\| \widetilde{A}^{(t-1)}(s, \cdot) \right\|_\infty \leq 1$ and Pinsker's inequality. Therefore,

$$\eta \left\langle \widetilde{A}^{(t-1)}(s, \cdot), \pi^*(\cdot \mid s) - \pi^{(t-1)}(\cdot \mid s) \right\rangle$$

$$\leq \mathrm{KL}\left(\pi^*(\cdot \mid s) \| \pi^{(t-1)}(\cdot \mid s)\right) - \mathrm{KL}\left(\pi^*(\cdot \mid s) \| \pi^{(t)}(\cdot \mid s)\right) + 2\eta^2 + \eta\beta\mathrm{KL}\left(\pi^*(\cdot \mid s) \| \pi^{\boldsymbol{\theta}^{\mathrm{ref}}}(\cdot \mid s)\right).$$

By telescoping, we have

$$\eta \sum_{t=1}^T \left\langle \widetilde{A}^{(t-1)}(s, \cdot), \pi^*(\cdot \mid s) - \pi^{(t-1)}(\cdot \mid s) \right\rangle$$

$$\leq \mathrm{KL}\left(\pi^*(\cdot \mid s) \| \pi^{(0)}(\cdot \mid s)\right) - \mathrm{KL}\left(\pi^*(\cdot \mid s) \| \pi^{(T)}(\cdot \mid s)\right) + 2\eta^2 T + \eta\beta\mathrm{KL}\left(\pi^*(\cdot \mid s) \| \pi^{\boldsymbol{\theta}^{\mathrm{ref}}}(\cdot \mid s)\right) T$$

$$\overset{(i)}{\leq} \mathrm{KL}\left(\pi^*(\cdot \mid s) \| \pi^{(0)}(\cdot \mid s)\right) + 2\eta^2 T + \eta\beta\mathrm{KL}\left(\pi^*(\cdot \mid s) \| \pi^{\boldsymbol{\theta}^{\mathrm{ref}}}(\cdot \mid s)\right) T.$$

$(i)$ uses the non-negativity of KL-divergence. By dividing $\eta$ on both sides, we have

$$\sum_{t=1}^T \left\langle \widetilde{A}^{(t-1)}(s, \cdot), \pi^*(\cdot \mid s) - \pi^{(t-1)}(\cdot \mid s) \right\rangle$$

$$\leq \frac{1}{\eta}\mathrm{KL}\left(\pi^*(\cdot \mid s) \| \pi^{(0)}(\cdot \mid s)\right) + 2\eta T + \beta\mathrm{KL}\left(\pi^*(\cdot \mid s) \| \pi^{\boldsymbol{\theta}^{\mathrm{ref}}}(\cdot \mid s)\right) T$$

$$\overset{(i)}{=} \frac{1}{\eta}\mathrm{KL}\left(\pi^*(\cdot \mid s) \| \pi^{\boldsymbol{\theta}^{\mathrm{ref}}}(\cdot \mid s)\right) + 2\eta T + \beta\mathrm{KL}\left(\pi^*(\cdot \mid s) \| \pi^{\boldsymbol{\theta}^{\mathrm{ref}}}(\cdot \mid s)\right) T.$$

(i) is because $\pi^{(0)}(\cdot \mid s) = \pi^{\boldsymbol{\theta}^{\mathrm{ref}}}(\cdot \mid s)$ by the initialization of Algorithm 2. $\qquad\square$

**Lemma E.4.** For any sequence of policies $\pi^{(1)}, \pi^{(2)}, \cdots, \pi^{(T)}$ and policy $\pi$, we have

$$\sum_{t=1}^T \left( V^\pi(s_{\mathrm{root}}) - V^{\pi^{(t)}}(s_{\mathrm{root}}) \right) = \sum_{s \in \mathcal{S} \backslash \mathcal{S}_H} \mu^\pi(s) \sum_{t=1}^T \left\langle Q^{\pi^{(t)}}(s, \cdot), \pi(\cdot \mid s) - \pi^{(t)}(\cdot \mid s) \right\rangle.$$

*Proof.* The lemma can be proved by induction. When $H = 1$, Lemma E.4 holds since $Q^{\pi^{(t)}}(s_{\text{root}}, a) = \mathcal{R}(\mathcal{T}(s_{\text{root}}, a)) = Q^{\pi}(s_{\text{root}}, a)$ for any action $a \in [B]$ and $t \in [T]$. Therefore,

$$\sum_{s \in \mathcal{S} \backslash \mathcal{S}_H} \sum_{t=1}^{T} \mu^{\pi}(s) \left\langle Q^{\pi^{(t)}}(s, \cdot), \pi(\cdot \mid s) - \pi^{(t)}(\cdot \mid s) \right\rangle$$

$$= \sum_{t=1}^{T} \mu^{\pi}(s_{\text{root}}) \left( \left\langle Q^{\pi}(s_{\text{root}}, \cdot), \pi(\cdot \mid s_{\text{root}}) \right\rangle - \left\langle Q^{\pi^{(t)}}(s_{\text{root}}, \cdot), \pi^{(t)}(\cdot \mid s_{\text{root}}) \right\rangle \right)$$

$$= \sum_{t=1}^{T} \left( V^{\pi}(s_{\text{root}}) - V^{\pi^{(t)}}(s_{\text{root}}) \right).$$

For any two nodes $s, s'$, we write $s \sqsubseteq s'$ if $s$ is an ancestor of $s'$ in the search tree. Consider when Lemma E.4 holds for any search tree of height $H \leq H_0$. Then, for $H = H_0 + 1$, we have

$$\sum_{s \in \mathcal{S} \backslash \mathcal{S}_H} \sum_{t=1}^{T} \mu^{\pi}(s) \left\langle Q^{\pi^{(t)}}(s, \cdot), \pi(\cdot \mid s) - \pi^{(t)}(\cdot \mid s) \right\rangle$$

$$= \sum_{t=1}^{T} \mu^{\pi}(s_{\text{root}}) \left\langle Q^{\pi^{(t)}}(s_{\text{root}}, \cdot), \pi(\cdot \mid s_{\text{root}}) - \pi^{(t)}(\cdot \mid s_{\text{root}}) \right\rangle$$

$$+ \sum_{a=1}^{B} \sum_{\substack{s \in \mathcal{S} \backslash \mathcal{S}_H : \\ \mathcal{T}(s_{\text{root}}, a) \sqsubseteq s}} \sum_{t=1}^{T} \mu^{\pi}(s) \left\langle Q^{\pi^{(t)}}(s, \cdot), \pi(\cdot \mid s) - \pi^{(t)}(\cdot \mid s) \right\rangle.$$

Then, according to the induction hypothesis, for any $a \in [B]$, since the subtree rooted at $\mathcal{T}(s_{\text{root}}, a)$ is a tree of height $H_0$, we have

$$\sum_{\substack{s \in \mathcal{S} \backslash \mathcal{S}_H : \\ \mathcal{T}(s_{\text{root}}, a) \sqsubseteq s}} \sum_{t=1}^{T} \mu^{\pi}(s) \left\langle Q^{\pi^{(t)}}(s, \cdot), \pi(\cdot \mid s) - \pi^{(t)}(\cdot \mid s) \right\rangle$$

$$= \pi(a \mid s_{\text{root}}) \sum_{t=1}^{T} \left( V^{\pi}(\mathcal{T}(s_{\text{root}}, a)) - V^{\pi^{(t)}}(\mathcal{T}(s_{\text{root}}, a)) \right).$$

Moreover, by definition, we have $Q^{\pi^{(t)}}(s_{\text{root}}, a) = V^{\pi^{(t)}}(\mathcal{T}(s_{\text{root}}, a))$. Therefore,

$$\sum_{s \in \mathcal{S} \backslash \mathcal{S}_H} \sum_{t=1}^{T} \mu^{\pi}(s) \left\langle Q^{\pi^{(t)}}(s, \cdot), \pi(\cdot \mid s) - \pi^{(t)}(\cdot \mid s) \right\rangle$$

$$= \sum_{t=1}^{T} \mu^{\pi}(s_{\text{root}}) \left\langle Q^{\pi^{(t)}}(s_{\text{root}}, \cdot), \pi(\cdot \mid s_{\text{root}}) - \pi^{(t)}(\cdot \mid s_{\text{root}}) \right\rangle$$

$$+ \sum_{a=1}^{B} \pi(a \mid s_{\text{root}}) \sum_{t=1}^{T} \left( V^{\pi}(\mathcal{T}(s_{\text{root}}, a)) - V^{\pi^{(t)}}(\mathcal{T}(s_{\text{root}}, a)) \right)$$

$$= \sum_{t=1}^{T} \sum_{a=1}^{B} \left( \pi(a \mid s_{\text{root}}) - \pi^{(t)}(a \mid s_{\text{root}}) \right) V^{\pi^{(t)}}(\mathcal{T}(s_{\text{root}}, a))$$

$$+ V^{\pi}(s_{\text{root}}) T - \sum_{a=1}^{B} \pi(a \mid s_{\text{root}}) \sum_{t=1}^{T} V^{\pi^{(t)}}(\mathcal{T}(s_{\text{root}}, a))$$

$$= \sum_{t=1}^{T} \left( V^{\pi}(s_{\text{root}}) - V^{\pi^{(t)}}(s_{\text{root}}) \right).$$

Therefore, Lemma E.4 also holds when $H = H_0 + 1$ and thus we can conclude the proof. $\square$

**Lemma E.6.** For any node $s \in \mathcal{S} \setminus \mathcal{S}_H$ and $t \in [T]$, we have

$$\eta \left\langle \widetilde{A}^{(t-1)}(s, \cdot), \pi^*(\cdot \mid s) - \pi^{(t)}(\cdot \mid s) \right\rangle$$

$$\leq \text{KL} \left( \pi^*(\cdot \mid s) \| \pi^{(t-1)}(\cdot \mid s) \right) - \text{KL} \left( \pi^*(\cdot \mid s) \| \pi^{(t)}(\cdot \mid s) \right) - \text{KL} \left( \pi^{(t)}(\cdot \mid s) \| \pi^{(t-1)}(\cdot \mid s) \right)$$

$$+ \eta \beta \text{KL} \left( \pi^*(\cdot \mid s) \| \pi^{\boldsymbol{\theta}^{\text{ref}}}(\cdot \mid s) \right).$$

*Proof.* Let $h$ be the height of $s$. There are three possibilities on $\nabla_{\pi(\cdot \mid s)} \mathcal{J}^{(t-1)}$: (I) $\widetilde{A}^{(t-1)}(s, \cdot) + \beta \log \pi^{(t-1)}(\cdot \mid s) - \beta \log \pi^{\boldsymbol{\theta}^{\text{ref}}}(\cdot \mid s) + \beta \mathbf{1}$; (II) A one-hot vector with only index $a_h^*$ be $\frac{\beta}{\pi^{(t-1)}(\cdot \mid s)}$; (III) $\mathbf{0}$.

Then, we will show that (D.2) is equivalent to the following in different cases.

**Lemma E.7.** For any $t \in \{1, 2, \cdots, T\}$, $h \in \{0, 1, \cdots, H-1\}$, and node $s \in \mathcal{S}_h$, (D.2) is equivalent to the following,

$$\pi^{(t)}(\cdot \mid s) = \underset{\pi(\cdot \mid s) \in \Delta^B}{\operatorname{argmin}} \left\langle -\widetilde{A}^{(t-1)}(s, \cdot), \pi(\cdot \mid s) \right\rangle + \beta \text{KL} \left( \pi(\cdot \mid s) \| \pi^{\boldsymbol{\theta}^{\text{ref}}}(\cdot \mid s) \right)$$

$$+ \frac{1}{\eta} \text{KL} \left( \pi(\cdot \mid s) \| \pi^{(t-1)}(\cdot \mid s) \right) \tag{I}$$

$$\pi^{(t)}(\cdot \mid s) = \underset{\pi(\cdot \mid s) \in \Delta^B}{\operatorname{argmin}} \left\langle -\nabla_{\pi(\cdot \mid s)} \mathcal{J}^{(t-1)}, \pi(\cdot \mid s) \right\rangle + \frac{1}{\eta} \text{KL} \left( \pi(\cdot \mid s) \| \pi^{(t-1)}(\cdot \mid s) \right), \tag{II, III}$$

where (I), (II), (III) stand for the cases when

$$\nabla_{\pi(\cdot \mid s)} \mathcal{J}^{(t-1)} = \begin{cases} \widetilde{A}^{(t-1)}(s, \cdot) + \beta \log \pi^{(t-1)}(\cdot \mid s) - \beta \log \pi^{\boldsymbol{\theta}^{\text{ref}}}(\cdot \mid s) + \beta \mathbf{1} & \text{(I)} \\ \text{A one-hot vector with only index } a_h^* \text{ be } \frac{\beta}{\pi^{(t-1)}(\cdot \mid s)} & \text{(II)} \\ \mathbf{0}. & \text{(III)} \end{cases}$$

Then, we will introduce a special case of Lemma 3.0.3 from Liu [2025].

**Lemma E.8.** For any node $s$, vector $\boldsymbol{g} \in \mathbb{R}^B$, $\eta > 0$, $\beta_0 \geq 0$, policy $\boldsymbol{x}^{(0)} \in \Delta^B$, and reference policy $\boldsymbol{x}^{\text{ref}} \in \Delta^B$, let

$$\boldsymbol{x}^{(1)} = \underset{\boldsymbol{x} \in \Delta^B}{\operatorname{argmin}} \left\{ \langle \boldsymbol{g}, \boldsymbol{x} \rangle + \beta_0 \text{KL} \left( \boldsymbol{x} \| \boldsymbol{x}^{\text{ref}} \right) + \frac{1}{\eta} \text{KL} \left( \boldsymbol{x} \| \boldsymbol{x}^{(0)} \right) \right\}.$$

Then, for any $\boldsymbol{x}^{(2)} \in \Delta^B$, we have

$$\eta \beta_0 \text{KL} \left( \boldsymbol{x}^{(1)} \| \boldsymbol{x}^{\text{ref}} \right) - \eta \beta_0 \text{KL} \left( \boldsymbol{x}^{(2)} \| \boldsymbol{x}^{\text{ref}} \right) + \eta \left\langle \boldsymbol{g}, \boldsymbol{x}^{(1)} - \boldsymbol{x}^{(2)} \right\rangle \tag{E.7}$$

$$\leq \text{KL} \left( \boldsymbol{x}^{(2)} \| \boldsymbol{x}^{(0)} \right) - (1 + \eta \beta_0) \text{KL} \left( \boldsymbol{x}^{(2)} \| \boldsymbol{x}^{(1)} \right) - \text{KL} \left( \boldsymbol{x}^{(1)} \| \boldsymbol{x}^{(0)} \right).$$

Consider (I) first. For any node $s \in \mathcal{S} \setminus \mathcal{S}_H$ and $t \in [T]$, by taking $\boldsymbol{x}^{(2)} = \pi^*(\cdot \mid s), \boldsymbol{x}^{(1)} = \pi^{(t)}(\cdot \mid s), \boldsymbol{x}^{(0)} = \pi^{(t-1)}(\cdot \mid s), \boldsymbol{x}^{\text{ref}} = \pi^{\boldsymbol{\theta}^{\text{ref}}}(\cdot \mid s), \boldsymbol{g} = -\widetilde{A}^{(t-1)}(s, \cdot)$ and $\beta_0 = \beta$, we have

$$\eta \beta \text{KL} \left( \pi^{(t)}(\cdot \mid s) \| \pi^{\boldsymbol{\theta}^{\text{ref}}}(\cdot \mid s) \right) - \eta \beta \text{KL} \left( \pi^*(\cdot \mid s) \| \pi^{\boldsymbol{\theta}^{\text{ref}}}(\cdot \mid s) \right)$$

$$+ \eta \left\langle \widetilde{A}^{(t-1)}(s, \cdot), \pi^*(\cdot \mid s) - \pi^{(t)}(\cdot \mid s) \right\rangle$$

$$\leq \text{KL} \left( \pi^*(\cdot \mid s) \| \pi^{(t-1)}(\cdot \mid s) \right) - (1 + \eta \beta) \text{KL} \left( \pi^*(\cdot \mid s) \| \pi^{(t)}(\cdot \mid s) \right) - \text{KL} \left( \pi^{(t)}(\cdot \mid s) \| \pi^{(t-1)}(\cdot \mid s) \right).$$

Further, by the non-negativity of KL-divergence, we have

$$\eta \left\langle \widetilde{A}^{(t-1)}(s, \cdot), \pi^*(\cdot \mid s) - \pi^{(t)}(\cdot \mid s) \right\rangle$$

$$\leq \text{KL} \left( \pi^*(\cdot \mid s) \| \pi^{(t-1)}(\cdot \mid s) \right) - \text{KL} \left( \pi^*(\cdot \mid s) \| \pi^{(t)}(\cdot \mid s) \right) - \text{KL} \left( \pi^{(t)}(\cdot \mid s) \| \pi^{(t-1)}(\cdot \mid s) \right)$$

$$+ \eta \beta \text{KL} \left( \pi^*(\cdot \mid s) \| \pi^{\boldsymbol{\theta}^{\text{ref}}}(\cdot \mid s) \right).$$

Consider (II). For any node $s \in \mathcal{S} \setminus \mathcal{S}_H$ and $t \in [T]$, by taking $\boldsymbol{x}^{(2)} = \pi^*(\cdot \mid s), \boldsymbol{x}^{(1)} = \pi^{(t)}(\cdot \mid s), \boldsymbol{x}^{(0)} = \pi^{(t-1)}(\cdot \mid s), \boldsymbol{x}^{\text{ref}} = \pi^{\boldsymbol{\theta}^{\text{ref}}}(\cdot \mid s), \boldsymbol{g} = -\nabla_{\pi(\cdot \mid s)} \mathcal{J}^{(t-1)}$ and $\beta_0 = 0$ in Lemma E.8, we have

$$\eta \left\langle \nabla_{\pi(\cdot \mid s)} \mathcal{J}^{(t-1)}, \pi^*(\cdot \mid s) - \pi^{(t)}(\cdot \mid s) \right\rangle$$

$$\leq \text{KL}\left(\pi^*(\cdot \mid s) \| \pi^{(t-1)}(\cdot \mid s)\right) - \text{KL}\left(\pi^*(\cdot \mid s) \| \pi^{(t)}(\cdot \mid s)\right) - \text{KL}\left(\pi^{(t)}(\cdot \mid s) \| \pi^{(t-1)}(\cdot \mid s)\right).$$

Moreover,

$$\left\langle \nabla_{\pi(\cdot \mid s)} \mathcal{J}^{(t-1)}, \pi^*(\cdot \mid s) - \pi^{(t)}(\cdot \mid s) \right\rangle = \beta \frac{\pi^*(a_h^* \mid s_h^*) - \pi^{(t)}(a_h^* \mid s_h^*)}{\pi^{(t-1)}(a_h^* \mid s_h^*)}$$

$$\overset{(i)}{\geq} 0$$

$$\overset{(ii)}{=} \left\langle \widetilde{A}^{(t-1)}(s, \cdot), \pi^*(\cdot \mid s) - \pi^{(t)}(\cdot \mid s) \right\rangle.$$

$(i)$ uses the fact that $\pi^*(a_h^* \mid s_h^*) = 1$ and $(ii)$ uses $\widetilde{A}^{(t-1)}(s, \cdot) = \boldsymbol{0}$ by definition. Therefore,

$$\left\langle \widetilde{A}^{(t-1)}(s, \cdot), \pi^*(\cdot \mid s) - \pi^{(t)}(\cdot \mid s) \right\rangle$$

$$\leq \text{KL}\left(\pi^*(\cdot \mid s) \| \pi^{(t-1)}(\cdot \mid s)\right) - \text{KL}\left(\pi^*(\cdot \mid s) \| \pi^{(t)}(\cdot \mid s)\right) - \text{KL}\left(\pi^{(t)}(\cdot \mid s) \| \pi^{(t-1)}(\cdot \mid s)\right). \quad \text{(E.8)}$$

For (III), which is $s$ off the sampled trajectory at step $t - 1$, by definition we have $\widetilde{A}^{(t-1)}(s, \cdot) = \boldsymbol{0}$. Then,

$$\left\langle \nabla_{\pi(\cdot \mid s)} \mathcal{J}^{(t-1)}, \pi^*(\cdot \mid s) - \pi^{(t)}(\cdot \mid s) \right\rangle = 0 = \left\langle \widetilde{A}^{(t-1)}(s, \cdot), \pi^*(\cdot \mid s) - \pi^{(t)}(\cdot \mid s) \right\rangle,$$

and (E.8) also holds. $\qquad \square$

**Lemma E.5.** For any $h \in \{0, 1, \cdots, H - 1\}$, we have

$$\text{KL}\left(\pi^*(\cdot \mid s_h^*) \| \pi^{\boldsymbol{\theta}^{\text{ref}}}(\cdot \mid s_h^*)\right) \leq \log B + 2 \left\| \boldsymbol{\theta}^{\text{ref}} \right\|_\infty.$$

*Proof.* For any $h \in \{0\} \cup [H - 1]$, since $\pi^*$ is deterministic, let $a_h^*$ be the action such that $\pi^*(a_h^* \mid s_h^*) = 1$. Then,

$$\text{KL}\left(\pi^*(\cdot \mid s_h^*) \| \pi^{\boldsymbol{\theta}^{\text{ref}}}(\cdot \mid s_h^*)\right) = \sum_{a=1}^B \pi^*(a \mid s_h^*) \log \frac{\pi^*(a \mid s_h^*)}{\pi^{\boldsymbol{\theta}^{\text{ref}}}(a \mid s_h^*)} = \log \frac{1}{\pi^{\boldsymbol{\theta}^{\text{ref}}}(a_h^* \mid s_h^*)}.$$

By definition, we have

$$\pi^{\boldsymbol{\theta}^{\text{ref}}}(a_h^* \mid s_h^*) = \frac{\exp\left(\theta^{\text{ref}}(s_h^*, a_h^*)\right)}{\sum_{a=1}^B \exp\left(\theta^{\text{ref}}(s_h^*, a)\right)} \geq \frac{\exp\left(-\left\|\boldsymbol{\theta}^{\text{ref}}\right\|_\infty\right)}{B \exp\left(\left\|\boldsymbol{\theta}^{\text{ref}}\right\|_\infty\right)} = \frac{\exp\left(-2\left\|\boldsymbol{\theta}^{\text{ref}}\right\|_\infty\right)}{B}.$$

Therefore,

$$\text{KL}\left(\pi^*(\cdot \mid s_h^*) \| \pi^{\boldsymbol{\theta}^{\text{ref}}}(\cdot \mid s_h^*)\right) \leq \log\left(B \cdot \exp\left(2\left\|\boldsymbol{\theta}^{\text{ref}}\right\|_\infty\right)\right) = \log B + 2\left\|\boldsymbol{\theta}^{\text{ref}}\right\|_\infty. \qquad \square$$

**Lemma E.7.** For any $t \in \{1, 2, \cdots, T\}$, $h \in \{0, 1, \cdots, H - 1\}$, and node $s \in \mathcal{S}_h$, (D.2) is equivalent to the following,

$$\pi^{(t)}(\cdot \mid s) = \operatorname*{argmin}_{\pi(\cdot \mid s) \in \Delta^B} \left\langle -\widetilde{A}^{(t-1)}(s, \cdot), \pi(\cdot \mid s) \right\rangle + \beta \text{KL}\left(\pi(\cdot \mid s) \| \pi^{\boldsymbol{\theta}^{\text{ref}}}(\cdot \mid s)\right)$$

$$+ \frac{1}{\eta} \text{KL}\left(\pi(\cdot \mid s) \| \pi^{(t-1)}(\cdot \mid s)\right) \qquad \text{(I)}$$

$$\pi^{(t)}(\cdot \mid s) = \operatorname*{argmin}_{\pi(\cdot \mid s) \in \Delta^B} \left\langle -\nabla_{\pi(\cdot \mid s)} \mathcal{J}^{(t-1)}, \pi(\cdot \mid s) \right\rangle + \frac{1}{\eta} \text{KL}\left(\pi(\cdot \mid s) \| \pi^{(t-1)}(\cdot \mid s)\right), \qquad \text{(II, III)}$$

where (I), (II), (III) stand for the cases when

$$\nabla_{\pi(\cdot \mid s)} \mathcal{J}^{(t-1)} = \begin{cases} \widetilde{A}^{(t-1)}(s, \cdot) + \beta \log \pi^{(t-1)}(\cdot \mid s) - \beta \log \pi^{\boldsymbol{\theta}^{\text{ref}}}(\cdot \mid s) + \beta \boldsymbol{1} & \text{(I)} \\ \text{A one-hot vector with only index } a_h^* \text{ be } \frac{\beta}{\pi^{(t-1)}(\cdot \mid s)} & \text{(II)} \\ \boldsymbol{0}. & \text{(III)} \end{cases}$$

*Proof.* The Lagrangian of $\left\langle -\widetilde{A}^{(t-1)}(s,\cdot), \pi(\cdot\,|\,s)\right\rangle + \beta\mathrm{KL}\left(\boldsymbol{x}\|\boldsymbol{x}^{\mathrm{ref}}\right) + \frac{1}{\eta}\mathrm{KL}\left(\pi(\cdot\,|\,s)\|\pi^{(t-1)}(\cdot\,|\,s)\right)$ is

$$\mathcal{L}\left(\pi^{(t)}(\cdot\,|\,s)\right) := \left\langle -\widetilde{A}^{(t-1)}(s,\cdot), \pi^{(t)}(\cdot\,|\,s)\right\rangle + \beta\mathrm{KL}\left(\pi^{(t)}(\cdot\,|\,s)\|\pi^{\boldsymbol{\theta}^{\mathrm{ref}}}(\cdot\,|\,s)\right)$$

$$+ \frac{1}{\eta}\mathrm{KL}\left(\pi^{(t)}(\cdot\,|\,s)\|\pi^{(t-1)}(\cdot\,|\,s)\right) + \lambda\left(\sum_{a=1}^{B}\pi^{(t)}(a\,|\,s) - 1\right).$$

For any action $a \in [B]$, by setting $\frac{\partial\mathcal{L}\left(\pi^{(t)}(\cdot\,|\,s)\right)}{\partial\pi^{(t)}(a\,|\,s)} = 0$, we have

$$-\widetilde{A}^{(t-1)}(s,\cdot) + \beta\log\left(\frac{\pi^{(t)}(a\,|\,s)}{\pi^{\boldsymbol{\theta}^{\mathrm{ref}}}(a\,|\,s)}\right) + \beta + \frac{1}{\eta}\log\left(\frac{\pi^{(t)}(a\,|\,s)}{\pi^{(t-1)}(a\,|\,s)}\right) + \frac{1}{\eta} + \lambda = 0,$$

which implies that $\pi^{(t)}(a\,|\,s) = \exp\left(\frac{-\eta\beta - 1 - \eta\lambda + \eta\widetilde{A}^{(t-1)}(s,\cdot) + \eta\beta\log\left(\pi^{\boldsymbol{\theta}^{\mathrm{ref}}}(a\,|\,s)\right) + \log\left(\pi^{(t-1)}(a\,|\,s)\right)}{1 + \eta\beta}\right)$.

By further setting $\frac{\partial\mathcal{L}\left(\pi^{(t)}(\cdot\,|\,s)\right)}{\partial\lambda} = 0$, we have

$$\sum_{a=1}^{B}\pi^{(t)}(a\,|\,s) = 1.$$

Therefore, by combining all pieces together, we have

$$\pi^{(t)}(a\,|\,s) \stackrel{(i)}{=} \exp\left(\frac{\eta\widetilde{A}^{(t-1)}(s,\cdot) + \eta\beta\log\left(\pi^{\boldsymbol{\theta}^{\mathrm{ref}}}(a\,|\,s)\right) + \log\left(\pi^{(t-1)}(a\,|\,s)\right)}{1 + \eta\beta}\right)/Z$$

$$\propto \exp\left(\frac{\eta\widetilde{A}^{(t-1)}(s,\cdot) + \eta\beta\log\left(\pi^{\boldsymbol{\theta}^{\mathrm{ref}}}(a\,|\,s)\right) + \log\left(\pi^{(t-1)}(a\,|\,s)\right)}{1 + \eta\beta}\right)$$

$$\propto \exp\left(\frac{\eta}{1 + \eta\beta}\widetilde{A}^{(t-1)}(s,\cdot) + \frac{\eta\beta}{1 + \eta\beta}\theta^{\mathrm{ref}}(s,a) + \frac{1}{1 + \eta\beta}\theta^{(t-1)}(s,a)\right).$$

In $(i)$, $Z = \sum_{a=1}^{B}\exp\left(\frac{\eta\widetilde{A}^{(t-1)}(s,\cdot) + \eta\beta\log\left(\pi^{\boldsymbol{\theta}^{\mathrm{ref}}}(a\,|\,s)\right) + \log\left(\pi^{(t-1)}(a\,|\,s)\right)}{1 + \eta\beta}\right)$.

For (II), (III), the proof can be concluded by setting $\beta = 0$ and changing $\widetilde{A}^{(t-1)}(s,\cdot)$ to $\nabla_{\pi(\cdot\,|\,s)}\mathcal{J}^{(t-1)}$. $\qquad\square$

