# OpenReview forum: "UFT: Unifying Supervised and Reinforcement Fine-Tuning"
_NeurIPS.cc/2025/Conference — NeurIPS 2025 poster_

### Official Review · Reviewer_N2ms · 2025-06-29

**Clarity:** 3
**Significance:** 4
**Originality:** 3
**Rating:** 5
**Confidence:** 5

**Summary:**

This paper proposes Unified Fine-Tuning (UFT), a novel post-training paradigm that combines traditional supervised fine-tuning (SFT) with reinforcement fine-tuning (RFT) in a single integrated process. The authors argue that SFT (learning from correct solutions) can lead to memorization/overfitting in large models, while RFT (learning via trial-and-error reward feedback) improves generalization but struggles when the base model is weak or rewards are sparse. UFT addresses these issues by blending supervised signals with reinforcement learning: during training, the model is guided with partial solutions (“hints”) to explore correct reasoning paths more frequently, and a hybrid objective optimizes both the reward of final answers and the log-likelihood of these hints. As training progresses, the hint length is gradually annealed to zero, smoothly transitioning from a fully supervised regime to pure reinforcement learning. The paper provides a theoretical analysis showing that UFT can break the exponential sample complexity bottleneck of standard RFT, achieving polynomial sample complexity in the length of the reasoning horizon. Empirically, UFT is evaluated on various reasoning-intensive tasks (arithmetic puzzles, math word problems, logic puzzles) across multiple model scales (0.5B to 3B parameters). The results demonstrate that UFT consistently outperforms both SFT and RFT alone, effectively obtaining the best of both: for smaller models UFT performs on par with SFT (which was superior to RFT in that regime), and for larger models UFT matches or exceeds RFT’s performance (which was superior to SFT for stronger models). Overall, the paper introduces a unified training framework that improves the reasoning capabilities of LLMs by combining the strengths of instruction-based supervised learning and reward-driven reinforcement learning in a principled way.

**Questions:**

1. How would UFT apply to open-ended tasks like dialogue or preference alignment, where step-by-step solutions aren’t available? Could model-generated hints or proxy signals be used? Clarifying this would help assess UFT’s broader applicability.

2. How critical are the hybrid loss and cosine hint annealing schedule? Would performance drop if we removed the supervised loss on hints or used a simpler schedule? Any ablation results would help clarify the importance of these components.

3. Can you clarify the assumptions behind Theorem 4.3? For instance, how realistic is the reward gap assumption, and do you use partial rewards in practice? Also, could UFT fail if the model becomes too dependent on hints?

4. How does UFT’s training cost compare to standard SFT or RFT? Does hint usage significantly increase sequence length or training time? Did UFT reach performance targets in fewer steps, as theory predicts? Any discussion on scalability would be useful.

**Ethical Concerns:**

["NO or VERY MINOR ethics concerns only"]

**Limitations:**

Partially discussed. UFT depends on high-quality intermediate steps, which may not exist in open-ended tasks. Broader applicability to instruction tuning or RLHF tasks is not demonstrated.

**Paper Formatting Concerns:**

No significant concerns. No sensitive data or risky deployment is involved.

**Quality:**

4

**Strengths And Weaknesses:**

Quality:

Strength: The paper’s technical quality is generally high. It proposes a clear training algorithm that integrates supervised and reinforcement signals, and backs it up with both theoretical guarantees and extensive experiments. The theoretical contribution is notable: the authors prove that in a long-horizon reasoning setting, pure RFT requires an exponential number of explorations in the worst case, whereas the UFT approach achieves a polynomial sample complexity – an exponential improvement in efficiency. This analysis (with formal theorems and proofs in the appendix) strengthens confidence that UFT’s performance gains are not just empirical luck but rooted in principle. The experimental evaluation is thorough: models of two different families (Qwen2.5 and Llama 3.2) and multiple sizes (0.5B, 1.5B, 3B) are fine-tuned on three distinct tasks. Results are consistently in favor of UFT, and the paper includes informative figures (e.g. training curves, accuracy vs. steps) illustrating how UFT avoids the pitfalls of each separate method (e.g. overcoming RFT’s stagnation on small models and SFT’s overfitting on large models).

Weaknesses: While the experiments support the claims, the scope of tasks and model scales is somewhat limited. The tasks (math puzzles, logic riddles) are all domains with a known ground-truth answer and a well-defined reward signal (answer correctness). It remains unclear how UFT would perform on more open-ended tasks or typical language model alignment settings (e.g. dialogue with human preference rewards), where “instruction tuning” usually refers to aligning with human instructions rather than solving puzzles. Additionally, the largest models tested are 3B parameters, which are relatively small by modern LLM standards – it would strengthen the paper to see if UFT’s benefits hold for truly large models (e.g. 7B, 13B or beyond) and on more diverse, real-world benchmarks. Another quality concern is that the method’s practicality relies on having high-quality solution trajectories (hints) for supervision; the authors used human-annotated step-by-step solutions from the dataset, but in general, obtaining or generating such intermediate solutions might be non-trivial for arbitrary tasks. The paper would benefit from a discussion or experiment on how noise or inaccuracies in the provided hints affect performance. Despite these points, the overall execution is solid – the algorithms are described in detail, and the claims are well-supported by evidence.

Strengths: The paper is clearly written and structured. The motivation is well-established with a relatable analogy (“learning by memorizing solutions vs. thinking through problems”), and background on SFT vs RFT is provided to orient the reader. The method is explained with both a conceptual figure and a concrete example of the UFT prompt format (including how the hint is inserted into the prompt), which makes the procedure easy to follow. Key components such as the hint-length annealing schedule and the combined loss function are described in a logical order. The inclusion of a “theoretical justification” section and an explicit list of contributions also aids clarity.

Weaknesses: A minor clarity issue is that Algorithm 1, which outlines the UFT procedure, is not discussed in much detail in the main text. A brief high-level explanation of its steps—such as how hints are sampled and how loss terms are scheduled—would help readers.

Significance:

 Strengths: This work tackles a timely and important problem in the fine-tuning of LLMs. As models grow and tasks demand reasoning, the limitations of purely supervised or purely RL-based fine-tuning have become apparent. A method that successfully combines the strengths of both has significant practical implications – it could lead to more data-efficient and capability-enhanced model training pipelines. The empirical gains demonstrated by UFT (outperforming SFT and RFT across tasks and scales) suggest that the approach can improve model reasoning without requiring larger models or prohibitively many RL training samples. The fact that UFT adapts to model capability (favoring memorization for weaker models and exploration for stronger ones) is particularly compelling – it hints that this single paradigm could simplify the fine-tuning strategy for a wide range of model sizes, rather than needing very different approaches for small vs. large LMs. The theoretical result is also significant in that it formally confirms an intuition about RL in long-horizon tasks and shows that providing intermediate feedback (hints) can dramatically accelerate learning.

Weakness:
The impact of UFT is somewhat limited by its reliance on curated intermediate solutions, which are more feasible for structured tasks like math but less available in broader domains like QA or dialogue. The chosen benchmarks, while suitable, are relatively narrow; showing gains on more widely-used or complex tasks would strengthen the significance. Still, the approach offers meaningful insights for improving LLM fine-tuning, with broader impact depending on future extensions.

Originality:

Strengths: The idea of unifying supervised and reinforcement fine-tuning is novel in the context of LLM training. Prior work has typically run SFT and RL sequentially (e.g. fine-tune on instructions, then apply RLHF), or chosen one based on context, but merging them into one framework with a seamless transition is a fresh approach. The use of partial solutions as hints effectively bridges the gap between “imitating correct reasoning” and “exploring new reasoning”, which is an original way to tackle the trade-off. The paper also introduces a smooth hint annealing technique (using a binomially sampled hint length that decays via cosine schedule) to avoid distributional shock – this is a thoughtful improvement over earlier “stagewise” or uniform sampling strategies (citing Xi et al., 2024) and appears to stabilize training. The theoretical analysis provided is another aspect of originality: it’s the first time (to my knowledge) that a unified training regime is proven to improve sample complexity for reasoning tasks, giving a new theoretical insight into why combining methods is beneficial.

Weaknesses: Some elements of UFT build on existing ideas, such as guiding RL with hints or mixing supervised and reinforcement learning, which makes the contribution more evolutionary than revolutionary. While the approach doesn’t introduce a fundamentally new paradigm, it offers a well-integrated and effective combination of known techniques. The novelty lies more in execution and analysis than in algorithmic invention, but the work remains sufficiently original for NeurIPS.

---

> ### Author Rebuttal · Authors · 2025-07-31
>
> # Response to Reviewer N2ms
>
> Thank you very much for your detailed and encouraging review. We greatly appreciate your recognition of both the theoretical and empirical contributions of UFT, as well as your thoughtful questions. Below, we address each of your points.
>
> ### Model scale and generalization to larger LLMs
>
> We fully agree that demonstrating UFT on larger models would further strengthen our claims. However, due to resource constraints (our current experiments already cost approximately $10,000), we were unable to scale up the experiments further.
>
> That said, our current results reveal key trends that may generalize. For instance, on Qwen2.5-1.5B:
>
> | Task (Qwen2.5-1.5B)| RFT | UFT|
> |------------------|---------|-----|
> | Countdown        |     71.48\% |71.48\%  |
> | MATH(3,4,5)      |   46.68\% | 47.95\% |
> | Logic            |    0.00\%  | 23.00\%   |
>
> We observe that RFT performs well on Countdown and MATH(3,4,5) but fails on Logic, where the model does not get enough knowledge through pretraining. UFT, by contrast, significantly outperforms RFT in this setting. We expect similar benefits on larger models when fine-tuning for extremely complex reasoning tasks or tasks requiring specific domain knowledge.
>
> ### Use of model-generated hints
>
> UFT is not limited to human-annotated hints. It only requires:
>
> - A reasoning trace. It can either be human annotations or chain-of-thoughts generated by other models.
> - A reward signal (*e.g.*, a rule-based reward model).
>
> The “hints” used in our work are not curated step-by-step annotations, but simply prefixes of existing solutions contained in the dataset. For each solution, we will divide it by sentence, and the hint length is the number of sentences. Here is an example from the MATH dataset:
>
> ```python
> [
> "For the piecewise function to be continuous, the cases must \"meet\" at $2$ and $-2$. ",
> "For example, $ax+3$ and $x-5$ must be equal when $x=2$. ",
> "This implies $a(2)+3=2-5$, which we solve to get $2a=-6 \\Rightarrow a=-3$. ",
> "Similarly, $x-5$ and $2x-b$ must be equal when $x=-2$. ",
> "Substituting, we get $-2-5=2(-2)-b$, which implies $b=3$. ",
> "So $a+b=-3+3=\\boxed{0}$."
> ]
> ```
>
> Therefore, this approach is easily extendable to LLM-generated traces or even imperfect intermediate outputs.
>
> In fact, prior work [3] has shown that LLM-generated explanations often outperform human-written ones for supervision. Recent work s1 [4] also utilized LLM-generated responses for fine-tuning. Hence, we expect UFT to perform even better when paired with high-quality model-generated responses.
>
> ### Description of the algorithm
>
> Thank you for highlighting this. We will update the main text to briefly explain Algorithm 1 at a high level, describing how hints are sampled and how the hybrid loss is scheduled during training.
>
> ### Ablation study on cosine annealing and supervised loss
>
> Figure 5 already includes an ablation where RFT is combined with cosine-annealed hints but without the supervised log-likelihood loss. The drop in performance in this setting demonstrates that the supervised loss on hints plays a key role in UFT’s effectiveness.
>
> ### Ablation on the hint schedule
>
> In Figure 5, we can see that $R^3$ (the hint length is sampled from a uniform distribution throughout the training) performs worse than RFT with cosine-annealed hints. Both of them adopt the standard RFT objective (3.1) without the log-likelihood on hints. This result also justifies the necessity of gradually annealing the hint length.
>
> On the other hand, the specific choice of hint length distribution is not important. To assess the sensitivity of UFT to this choice, we also experimented with a two-point distribution: for any expected hint length $p\cdot L\in [n, n+1)$, we set $l=n$ with probability $n+1-p\cdot L$ and $l=n+1$ with probability $p\cdot L-n$. $p$ is still determined by the cosine-annealing scheduler. Using this scheme, the final accuracy of Qwen2.5-0.5B (Countdown) is:
>
> - Two-point: 48.63\%
> - Binomial distribution: 50.39%
>
> This suggests that the choice of distribution's impact on overall performance is relatively small.
>
> ### Clarifications on Theorem 4.3
>
> For any rule-based reward model, the assumption holds because there is always a gap between the reward for the correct answer and the incorrect answer. For instance, Deepseek R1 [1] and DAPO [2].
>
> Regarding “partial rewards,” if this refers to assigning intermediate rewards (e.g., for correct reasoning steps), we currently do not use partial rewards in our setup. All rewards are terminal (binary correctness). However, extending UFT to settings with partial rewards is a promising direction for future work.
>
> ### Avoiding overreliance on hints
>
> Indeed, overreliance on hints is a real concern, and the cosine annealing strategy is specifically designed to mitigate this. As training progresses, the expected hint length smoothly decays, ensuring that the model learns to reason without external help, aligning training with the final evaluation distribution (no hints at inference time).
>
> ### Computation time
>
> UFT is faster than RFT per training step. Here is the training time per step averaged across tasks:
>
> | Model | RFT | UFT |
> |-------|-----|-----|
> | Qwen2.5-0.5B | 63.62 | 57.18 |
> | Qwen2.5-1.5B | 98.25 | 89.21 |
> | Qwen2.5-3B | 127.14 | 118.36 |
> | Llama-3.2-1B | 96.61 | 74.63 |
> | Llama-3.2-3B | 126.90 | 108.18 |
> | Average | 102.50 | 89.51 |
>
> This improvement stems from shorter generation lengths during the hint phase. We also report the rollout time per step below, showing clear gains:
>
> | Model | RFT | UFT |
> |-------|-----|-----|
> | Qwen2.5-0.5B | 26.75 | 20.47 |
> | Qwen2.5-1.5B | 41.34 | 32.40 |
> | Qwen2.5-3B | 63.98 | 55.35 |
> | Llama-3.2-1B | 52.30 | 32.13 |
> | Llama-3.2-3B | 65.22 | 48.28 |
> | Average | 49.92 | 37.72 |
>
> While UFT introduces a hybrid loss, this overhead is offset by a more efficient trajectory rollout procedure.
>
> ### References
>
> [1] Guo D, Yang D, Zhang H, et al. Deepseek-r1: Incentivizing reasoning capability in llms via reinforcement learning[J]. arXiv preprint arXiv:2501.12948, 2025.
>
> [2] Yu Q, Zhang Z, Zhu R, et al. Dapo: An open-source llm reinforcement learning system at scale[J]. arXiv preprint arXiv:2503.14476, 2025.
>
> [3] Ren X, Wu B, Liu L. I learn better if you speak my language: Understanding the superior performance of fine-tuning large language models with LLM-generated responses[J]. arXiv preprint arXiv:2402.11192, 2024.
>
> [4] Muennighoff N, Yang Z, Shi W, et al. s1: Simple test-time scaling[J]. arXiv preprint arXiv:2501.19393, 2025.
>
> [5] Agarwal A, Kakade S M, Lee J D, et al. On the theory of policy gradient methods: Optimality, approximation, and distribution shift[J]. Journal of Machine Learning Research, 2021, 22(98): 1-76.
>
> [6] Jin C, Liu Q, Wang Y, et al. V-Learning--A Simple, Efficient, Decentralized Algorithm for Multiagent RL[J]. arXiv preprint arXiv:2110.14555, 2021.

---

### Official Review · Reviewer_BALi · 2025-06-30

**Clarity:** 2
**Significance:** 2
**Originality:** 3
**Rating:** 4
**Confidence:** 4

**Summary:**

This paper proposes UFT, a post-training method that unifies SFT and RFT into a single integrated process. The authors argue that SFT suffers from overfitting while RFT depends heavily on base model strength, and UFT addresses both limitations by using partial solutions as "hints" to guide exploration while incorporating supervision signals. They provide theoretical analysis claiming exponential sample complexity improvements over standard RFT and empirical validation on reasoning tasks using various language models.

**Questions:**

- Hint length is sampled from a Binomial distribution. What is the rationale for this specific distributional choice? How would performance be affected by using a simpler scheme?
- Figure 4 (right) indicates that hints are phased out entirely after 300 steps. Does this imply the hint mechanism's primary role is as a warm-up, with the final performance advantage stemming solely from the modified objective function?
- The objective in Eq3.2 includes the term $KL(\pi^\ast\|\|\pi)$, which you estimate via a single monte carlo trace using the annotated trajectory $(s_h^\ast, a_h^\ast)$. This seems to be an exceptionally high-variance estimator. How does this not introduce significant instability during training?
- Could you provide a detailed ablation study that isolates the performance contribution of the $\sum KL(\pi^\ast\|\|\pi)$ term? Specifically, what is the performance of UFT if this term is removed, leaving only the GRPO objective and the standard log-likelihood on the hint?
- How do you justify the direct application of theoretical results derived from a deterministic, uniform tree-search model to the stochastic and unstructured nature of LLM text generation?
- The informal statement of Theorem 4.3 requires $\beta$ be "small enough." Could you formalize this condition and comment on whether the empirically optimal value of $\beta$ satisfies this theoretical constraint?

**Ethical Concerns:**

["NO or VERY MINOR ethics concerns only"]

**Final Justification:**

All my questions and concerns are resolved.

**Limitations:**

yes

**Quality:**

3

**Strengths And Weaknesses:**

pros

- The approach tries to combine SFT and RFT in a principled manner, which is an interesting topic.
- The paper provides clear motivation.
- The paper is well written.

cons

- I'm not fully convinced by the theoretical analysis. See the questions below.

---

> ### Author Rebuttal · Authors · 2025-07-30
>
> # Response to Reviewer BALi
>
> Thank you for your thoughtful and detailed feedback. We sincerely appreciate your insights, and we address your comments point by point below.
>
> ### Ablation study on distributions
>
> We chose the binomial distribution for hint length sampling due to its simplicity and desirable properties: (i) it is discrete with support over integers (hint lengths), and (ii) the expected hint length can be precisely controlled, which is crucial for implementing the cosine-annealing schedule.
>
> To assess the sensitivity of UFT to this choice, we also experimented with a two-point distribution: for any expected hint length $p\cdot L\in [n, n+1)$, we set $l=n$ with probability $n+1-p\cdot L$ and $l=n+1$ with probability $p\cdot L-n$. $p$ is still determined by the cosine-annealing scheduler. Using this scheme, the final accuracy of Qwen2.5-0.5B (Countdown) is:
>
> - Two-point: 48.63\%
> - Binomial distribution: 50.39%
>
> This suggests that the choice of distribution's impact on overall performance is relatively small.
>
> ### KL-Divergence estimation in (3.2)
>
> As we discussed in Line 180-186, $KL(\pi^{\*}|| \pi)$ (red term) is equivalent to the log-likelihood on the hint (the blue term in Eq 3.3) by definition of the KL-divergence.
>
> Specifically, $KL(\pi^{\*}|| \pi)=\sum_a \pi^{\*}(a|s)\log \frac{\pi^{\*}(a|s)}{\pi(a|s)}=\sum_a \pi^{\*}(a|s)\log \pi^{\*}(a|s) + \sum_a \pi^{\*}(a|s)\log \frac{1}{\pi(a|s)}$.
>
> Since only $\sum_a \pi^{\*}(a|s)\log \frac{1}{\pi(a|s)}$ is related to our policy $\pi$, minimizing the KL is equivalent to minimizing $\sum_a \pi^{\*}(a|s)\log \frac{1}{\pi(a|s)}$. Lastly, the log-likelihood on the hint, $\log \frac{1}{\pi(a|s)}$ with $a\sim \pi^{\*}(\cdot|s)$, is an unbiased estimator of $\sum_a \pi^{\*}(a|s)\log \frac{1}{\pi(a|s)}$.
>
> ### Performance gain
>
> The objective function of UFT is equivalent to that of RFT (GRPO) when $\sum KL(\pi^{\*}||\pi)$ is removed (c.f. Equation (3.1) and (3.2)). The only difference after removing $\sum KL(\pi^{\*}||\pi)$ is that UFT starts reasoning from the hint, while RFT starts from scratch.
>
> In Figure 5, we did an ablation study on using hint but **with/without** the modified objective function (the additional log-likelihood on hint, equivalent to the KL as stated above). It shows that the additional log-likelihood contributes to the final performance gain and helps the model to learn new knowledge.
>
> ### Implication of theory.
>
> Modeling LLM text generation as a uniform tree (each node representing the generation of the next token) is a standard and widely used abstraction in prior work [1][2].
>
> For LLM text generation, if we take each branch as the next token, then the whole text generation forms a path in the tree. Since the token table is fixed for LLMs, the branching factor $B$ is the same for all nodes. In other words, the search tree is uniform.
>
> Finally, our theoretical results serve to justify that integrating the supervised signal during RL helps improve sample complexity, as noted by Reviewer N2ms. While our analysis assumes uniform trees, the results can extend to imbalanced trees with minor adjustments. For instance, the lower bound in Theorem 4.2 can be understood more generally as requiring exploration of at least $\frac{\text{Number of Leaves}}{4}$ to achieve 50\% accuracy.
>
> ### $\beta$ value
>
> We acknowledge the reviewer’s request for a more detailed explanation regarding $\beta$. The formal version of Theorem 4.3, provided in Appendix E, shows that $\beta$ should satisfy: $\beta\leq O(\frac{\Delta}{(H+1)^2\log B})$.
>
> This is a conservative upper bound that holds under worst-case for arbitrary reference policies $\pi^{ref}$. A tighter (and more practical) condition depends on the KL divergence between the and the reference policy, which is $O(\frac{\Delta}{(H+1) KL(\pi||\pi^{ref})})$.
>
> In practice, due to pretraining, $\pi^{ref}$ might be close to $\pi^{\*}$, making it feasible to choose a larger $\beta$. In our experiments, we choose $\beta=0.001$, which satisfies the bound when $\pi^{ref}$ is not far from $\pi^{\*}$.
>
> ### References
>
> [1] Xie Y, Goyal A, Zheng W, et al. Monte carlo tree search boosts reasoning via iterative preference learning[J]. arXiv preprint arXiv:2405.00451, 2024.
>
> [2] Chen G, Liao M, Li C, et al. Alphamath almost zero: process supervision without process[J]. Advances in Neural Information Processing Systems, 2024, 37: 27689-27724.

---

> > ### Comment · Reviewer_BALi · 2025-08-04
> >
> > Thanks for your clarifications. Most of my questions have been resolved. I will continue to follow the discussions and feedback from other reviewers to decide my final score.

---

### Official Review · Reviewer_MekW · 2025-07-03

**Clarity:** 3
**Significance:** 3
**Originality:** 3
**Rating:** 5
**Confidence:** 4

**Summary:**

This paper introduces Unified Fine-Tuning (UFT), a novel post-training paradigm that integrates supervised fine-tuning (SFT) and reinforcement fine-tuning (RFT) into a single process. The authors address the fundamental trade-off in LLM post-training: SFT excels at knowledge acquisition but suffers from overfitting, while RFT provides better generalization but struggles with sparse rewards and depends heavily on base model capabilities. UFT resolves this by using gradually shortened "hints" and a hybrid objective function that combines reinforcement learning with log-likelihood maximization on these hints.

**Questions:**

1. While UFT demonstrates superior performance on most benchmarks, SFT-RFT occasionally outperforms UFT on certain tasks (e.g., Countdown and Logic), and Table 2 reveals that UFT doesn't universally dominate across all evaluation metrics. This raises critical questions about when practitioners should choose SFT-RFT versus UFT. What task characteristics, model properties, or dataset features determine the optimal choice between these methods?
2. The extensive mathematical formulation in lines 88-101 introduces numerous tree-based definitions and notations that appear underutilized in subsequent theoretical development. Notably, the core objective functions in Equations 3.1 and 3.2 seem largely independent of the elaborate tree structure definitions.
3.  The paper lacks clear guidance on how to strategically allocate training data between SFT and RFT phases. Should practitioners prioritize difficult examples for the SFT stage to provide strong foundational knowledge, or should simpler examples be used for supervised learning while reserving complex cases for reinforcement learning?
4. A comprehensive analysis comparing computational overhead between UFT, pure SFT, and pure RFT is notably absent.
5. The decision to begin zero-hint training at step 300 in Figure 4 appears empirically motivated rather than theoretically grounded. How can the optimal timing for beginning zero-hint training be determined across different model sizes, task complexities, and dataset characteristics?
6. Several theoretical claims lack sufficient justification, most notably the assertion on line 185 that a particular estimator is unbiased without providing formal proof. While the authors present extensive theoretical analysis in the appendix, key intermediate results like the optimal training step count remain underdeveloped.
7. The evaluation is primarily concentrated on mathematical and logical reasoning tasks using relatively small language models (≤3B parameters). This narrow scope limits the generalizability of findings to broader reasoning domains, or larger model scales.
8. Algorithm 2's Q-value computation (line 7) appears oversimplified. Additionally, unclear definitions of key metrics (e.g., "Average Score" in Figure 4) and missing comparative analyses (e.g., standard RFT performance versus RFT with cosine annealing in Figure 5) limit the comprehensiveness of the experimental evaluation.

**Ethical Concerns:**

["NO or VERY MINOR ethics concerns only"]

**Final Justification:**

After careful consideration of the rebuttal and discussions, I recommend accepting this paper. The authors have successfully addressed my primary concerns through comprehensive experimental evidence: (1) computational cost analysis showing UFT is actually faster than RFT due to shorter generation lengths during the hint phase, (2) extensive ablation studies demonstrating robustness to hyperparameter choices, and (3) clear practical value in providing a reliable unified approach when the optimal training strategy is unknown beforehand. While some limitations remain—particularly the evaluation scope being limited to mathematical/logical reasoning tasks on smaller models (≤3B) and the lack of theoretical convergence guarantees—the core contribution of seamlessly integrating SFT and RFT through gradually shortened hints represents a meaningful advance in LLM post-training. The method's consistent performance across diverse settings, especially avoiding catastrophic failures like RFT's 0% on Logic tasks, combined with its computational efficiency, makes this a valuable contribution to the community that merits publication.

**Limitations:**

Yes. However, there are only one limitation about human-annotated solutions. I suggest the author could add more limitation analysis and training cost of UFT.

**Quality:**

3

**Strengths And Weaknesses:**

## Strength
1. The paper addresses a genuine limitation in current fine-tuning approaches. The observation that SFT works well for small models but may lead to overfitting for larger ones, while RFT generalizes better but depends heavily on base model capabilities, is well-supported and practically relevant.
2. The core idea is intuitive. The cosine annealing schedule for hint length provides a smooth transition from supervised to reinforcement learning, avoiding the distribution shifts that plague staged approaches.
3. The evaluation across multiple model sizes (0.5B-3B), architectures (Qwen2.5, Llama-3.2), and tasks (Countdown, MATH, Logic puzzles) demonstrates the generality of the approach.

## Weakness
1.  The paper doesn't discuss the computational cost of UFT relative to baselines. The hybrid objective and hint processing likely increase training time.
2. The method introduces additional hyperparameters (hint length schedule, β weighting). While the paper shows UFT is effective, more analysis of sensitivity to these choices would help practitioners.
3. It would strengthen the paper to include other domains like code generation, scientific questions, or other reasoning domains to demonstrate broader applicability.

---

> ### Author Rebuttal · Authors · 2025-07-30
>
> # Response to Reviewer MekW
>
> Thank you for your comprehensive and thoughtful review. We deeply appreciate your questions and suggestions. Below, we address each of your points in detail.
>
> ### When to use UFT
>
> UFT’s main advantage lies in its robustness across tasks and model capacities. While SFT, RFT, or SFT-RFT may perform slightly better in specific scenarios, it is often difficult to determine the optimal fine-tuning strategy beforehand. UFT offers a strong, consistent alternative that adapts across regimes.
>
> As demonstrated in the table below (Qwen2.5-1.5B), RFT surpasses SFT-RFT on Countdown and MATH(3,4,5), but fails entirely on Logic. UFT, by contrast, maintains competitive or superior performance across all tasks:
>
> Task (Qwen2.5-1.5B)| SFT-RFT | RFT | UFT|
> |------------------|---------|-----|----|
> | Countdown        |   57.13\%      |   71.48\% |71.48\%  |
> | MATH(3,4,5)      |   36.91\%      |   46.68\% | 47.95\% |
> | Logic            |   29.6\%      |   0.00\%  | 23.00\%   |
> | Average          |   41.21\%    |   39.39\%    |   47.48 |
>
> This illustrates that while task complexity and model capacity are important, their interaction is nontrivial. UFT offers a reliable choice when the best training scheme is unknown.
>
> ### Computational cost comparison
>
> We evaluated the average training step time across models. Surprisingly, UFT is faster than RFT, due to shorter outputs in the early training phase (thanks to hints):
>
> | Model          | RFT    | UFT    |
> |---------------|---------|--------|
> | Qwen2.5-0.5B       | 63.62  | 57.18  |
> | Qwen2.5-1.5B          | 98.25  | 89.21  |
> | Qwen2.5-3B           | 127.14 | 118.36 |
> | Llama-3.2-1B    | 96.61  | 74.63  |
> | Llama-3.2-3B     | 126.90 | 108.18 |
> | Average            | 102.50  | 89.51   |
>
> This improvement stems from shorter generation lengths during the hint phase. We also report the rollout time per step below, showing clear gains:
>
> | Model | RFT | UFT |
> |-------|-----|-----|
> | Qwen2.5-0.5B | 26.75 | 20.47 |
> | Qwen2.5-1.5B | 41.34 | 32.40 |
> | Qwen2.5-3B | 63.98 | 55.35 |
> | Llama-3.2-1B | 52.30 | 32.13 |
> | Llama-3.2-3B | 65.22 | 48.28 |
> | Average | 49.92 | 37.72 |
>
> Pure SFT remains the most efficient (about 6-9x faster than RFT/UFT), as it does not involve trajectory sampling or reward computation.
>
> ### Hint length ablation
>
> We conducted a study on Qwen2.5-0.5B (MATH(3,4,5)), testing dividing the solution into $L=4/5/6$ pieces uniformly and sampling hint length among $\{0,1,\dots, L\}$. We choose MATH(3,4,5) instead of Countdown because the solution length of MATH(3,4,5) is relatively longer.
>
> - Hint length $L=4$: 27.44\%
> - Hint length $L=5$: 27.15\%
> - Hint length $L=6$: 25.20\%
>
> This suggests that UFT is relatively insensitive to the total number of pieces we divided the solution into.
>
> ### $\beta$ ablation
>
> We evaluated different $\beta$ values on Qwen2.5-0.5B (Countdown):
>
> - $\beta=0.0005$: 46.09\%
> - $\beta=0.001$: 50.39\%
> - $\beta=0.002$: 59.95\%
>
> A higher $\beta$ amplifies the impact of the supervised log-likelihood term, helping the model learn more from hints. For all main experiments, we adopt $\beta$ = 0.001, the default in VERL. For larger models, our preliminary results show that varying $\beta$ yields minimal performance changes, likely due to their stronger pretrained priors. Here are the results of Qwen2.5-3B (Countdown).
>
> - $\beta=0.0005$: 77.93\%
> - $\beta=0.001$: 75.59\%
> - $\beta=0.002$: 78.22\%
>
> ### Length of hint phase ablation
>
> We tested different durations for the hint phase on Qwen2.5-0.5B (Countdown):
>
> - $T_{hint}=200$: 52.15\%
> - $T_{hint}=300$: 50.39\%
> - $T_{hint}=400$: 50.00\%
>
> These small variations confirm that UFT is robust to the hint phase length.
>
> ### Data allocation strategy
>
> We appreciate your insightful suggestion. We agree that placing harder examples in the supervised phase and easier ones in the RL phase may further enhance performance. Moreover, adaptively adjusting hint length based on the hardness of each data point is also promising.
>
> However, the current paper’s goal is to demonstrate the value of integrating supervising signals to reinforcement learning, and we leave adaptive data allocation to future work.
>
> ### Clarification of theory
>
> - Tree Structure (Lines 82–101): These definitions are necessary to formally define the trajectory sampling and expected return in Section 3.
> - Optimal Step Count: Since GRPO is a convergent algorithm, the more steps we train, the higher the final accuracy is. The accurate value of $T$ of achieving $50\%$ accuracy is given in Lemma E.1.
> - In Algorithm 2, using reward as the Q-value follows the practice in GRPO [1], where advantage is computed as $\tilde A(s,a)=\frac{r(s,a)-mean({\bf r})}{std({\bf r})}$. Our implementation omits normalization for simplicity, in line with GRPO variants [2].
>
> ### References
>
> [1] Shao Z, Wang P, Zhu Q, et al. Deepseekmath: Pushing the limits of mathematical reasoning in open language models[J]. arXiv preprint arXiv:2402.03300, 2024.
>
> [2] Liu Z, Chen C, Li W, et al. Understanding r1-zero-like training: A critical perspective[J]. arXiv preprint arXiv:2503.20783, 2025.

---

> ### Comment · Reviewer_MekW · 2025-08-06
>
> Thank you for your patient and detailed response, which has addressed most of my concerns. The computational cost analysis showing UFT's efficiency advantage over RFT is particularly insightful, and the ablation studies demonstrate the method's robustness across different hyperparameter settings. I look forward to seeing UFT's performance on larger models. In recognition of the authors' thorough clarifications and the value of their contribution, I am raising my score.

---

> > ### Author Response · Authors · 2025-08-07
> >
> > Thank you for your thoughtful response and for raising your score. We're glad the analyses and ablations were helpful.

---

### Official Review · Reviewer_KErd · 2025-07-04

**Clarity:** 3
**Significance:** 4
**Originality:** 3
**Rating:** 4
**Confidence:** 3

**Summary:**

This paper brings together supervised and reinforcement fine tuning into a single framework. The authors claim that the combination of SFT and RFT can actually break the exponential sample complexity. SFT+RFT hold promise of combining curriculum memorization with reward-driven exploration, thereby enabling thinking/reasoning and memorizing at the same time. The authors contrast it with approaches that use SFT or RFT alone, or combine them sequentially.

**Questions:**

* Does UFT unification proposed by authors generalize to larger models that have enough capacity to just train on RFT alone?
* My main concern is lack of in-depth experimentation with a broader set of models and larger model sizes (e.g. 70B). If the authors convince me that UFT generalizes to more and larger models -- this can be convincing evidence of desired generalization (and could help +1 my score). Otherwise, UFT's contributions might be very constrained.
* it would be good to show how UFT compares to larger SOTA models given it's better convergence and sample efficiency (i.e. within some time budget or upper limit on the number of steps). While larger SOTA models may eventually converge to better accuracy, UFT may offer competitive performance within a time or step limit.

**Ethical Concerns:**

["NO or VERY MINOR ethics concerns only"]

**Limitations:**

yes

**Quality:**

3

**Strengths And Weaknesses:**

**Strengths**
* smooth transition from SFT to RFT during fine tuning is intuitive, making the exploration task progressively harder and reducing the dependency on supervision gradually over time.
* The authors consider previous work thoroughly, citing R^3
* The paper offers intuitive motivation for unifying SFT with RFT based on human intelligence (though some may find drawing these parallels as a disadvantage)
* theoretical proof is offered for polynomial sample complexity for UFT.
* information-theoretic intuition/explanation is offered (sec 3.2) for why RFT's sparse reward signal (correct/incorrect) is inferior to injecting new knowledge through partial SFT hints. Thus, UFT is shown to leverage the strengths of both fine tuning approaches.
* UFT reduces to SFT as a special case, mathematically (remark 3.1)

**Weaknesses**
* Exploration was done on non-SOTA models that are quite small Qwen2.5-0.5/1.5/3B. (only an ablation study for hint length predictors was done on LLama 3.2-1)
* UFT may not generalize (or not offer any benefits) for larger reasoning models.

---

> ### Author Rebuttal · Authors · 2025-07-30
>
> # Response to Reviewer KErd
>
> We sincerely appreciate your thoughtful review and constructive feedback. Below, we address your primary concern regarding the generalization of UFT to larger models and large-scale experiments.
>
> ### UFT's generalization to larger models
>
> Even though we cannot afford training a 70B model, we believe UFT will be helpful for 70B models on tasks out of its capacity (unseen tasks during pretraining / extremely hard reasoning problems).
>
> While we are currently unable to run experiments on 70B-scale models due to computational constraints, we believe that UFT will be beneficial even for such models, especially on tasks that exceed their pretraining capabilities. For instance, some tasks involving specific domain knowledge or extremely complex reasoning problems.
>
> The following empirical results may hint at this. For instance, on the logic puzzle, which requires more abstract reasoning, Qwen2.5-1.5B performs poorly with RFT alone (0.00%) but achieves 23.00% with UFT:
>
> Task (Qwen2.5-1.5B)|  RFT | UFT|
> |------------------|-----|----|
> | Countdown        |    71.48\% |71.48\%  |
> | MATH(3,4,5)      |      46.68\% | 47.95\% |
> | Logic            |     0.00\%  | 23.00\%   |
>
> In conclusion, although Qwen2.5-1.5B has enough capacity for Countdown and MATH(3,4,5), it is insufficient in solving the logic puzzle and UFT outperforms RFT by a large margin. Therefore, it is likely that for larger models, UFT will be helpful when solving extremely hard tasks.
>
> This stark difference in Logic suggests that even when a model appears to have “enough capacity” (as Qwen2.5-1.5B does for Countdown and MATH), it can still benefit significantly from UFT when the task challenges its reasoning ability. Hence, for much larger models facing extremely complex tasks, we anticipate that UFT will continue to provide meaningful advantages.
>
> ### Experiments on large models
>
> We fully agree that broader experiments would offer more insight. However, as noted in Appendix B.2, our current experiments already incurred a cost of approximately $10,000, and scaling to 70B models would be prohibitively expensive for us at this stage.
>
> That said, we emphasize that absolute model size is not the sole driver of UFT's benefits. What matters more is the relative complexity between the model's capacity and the task. Our findings show that UFT outperforms RFT in settings relatively hard for the model (such as the logic puzzle for Qwen2.5-1.5B), which is a likely scenario even for large models on particularly difficult or unfamiliar problems.

---

> ### Comment · Reviewer_KErd · 2025-08-08
> **addressing concern on experimental setup**
>
> I definitely appreciate the significant cost of experimentation. The authors have partially addressed my concern with their argument whereby Qwen2.5-1.5B has sufficient capacity for countdown and MATH but fails on Logic when RFT is used. I buy that. However, instead of having to rely on this argument without further experimental evidence, it could be better/easier for the authors to simply limit the scope and positioning of this paper to SLM models. There's no reason to make an argument for larger models if you can't validate it with experiments. You can simply argue that SLMs are important for edge compute cases, and your proposed UFT outperforms SOTA on SLMs. I would +1 the score to Accept given such an argument. But it is much harder to agree to your claim that UFT is more generally applicable (plainly because generalizability to LLMs is not substantiated, regardless of the reason). My score was already positive to begin with, so I'm keeping it the same. I would assign a higher score either if you (a) limited the scope to SLMs or if you (b) provided experimental validation of claims of UFT generality for LLMs. I hope my rationale makes sense. It's good work -- as reflected in my extensive list of strengths. Thank you.

---

> > ### Author Response · Authors · 2025-08-08
> >
> > Thank you very much for your constructive feedback and positive assessment of our work. We understand your point about the scope of claims and the need for experimental validation.
> >
> > In future versions, we will state in the Conclusion and Limitations that our experiments are not conducted on SOTA models (*e.g.*, 70B models) due to computational cost. We believe this will address your concern and align the paper’s positioning with our evidence.
> >
> > We appreciate your clear guidance and recognition of our work!

---

### Note · Authors · 2025-08-13

Dear AC and Reviewers,

We sincerely thank you for your time and effort during the discussion.

Reviewers MekW and BALi raised their scores after our rebuttal, and N2ms maintained an Accept. Reviewer KErd also expressed a positive view and stated they would consider raising the score if we clarified the scope of our experiments. In our latest response, we committed to clarifying this in future versions, and would be grateful if **KErd might revisit the rating**.

Reviewer **KErd** acknowledged that our paper *"consider previous work thoroughly"* and *"offers intuitive motivation"*. KErd’s main concern was that our experiments were not conducted on SOTA models (*e.g.*, 70B). We provided empirical evidence suggesting that UFT would benefit SOTA models on tasks beyond their pretraining capacity (unseen tasks or extremely hard reasoning problems). KErd observed that we had partially addressed the concern and recommended clarifying the scope of our experiments. We appreciate this suggestion and will incorporate the clarification in future versions.

Reviewer **MekW** highlighted that *"the evaluation across multiple ... demonstrates the **generality**"* of our approach, but raised questions about UFT’s computational cost and sensitivity to hyperparameters. Then, we demonstrated that UFT is faster than RFT, as providing a hint leads to shorter responses and thus reduced rollout time. We also presented ablation studies on hyperparameters.

Following this, MekW stated that *"the computational cost analysis ... is particularly insightful, and the ablation studies demonstrate the method’s **robustness** across different hyperparameter settings"*, and **raised the rating**.

Reviewer **BALi** noted that *"the paper provides clear motivation"* and *"... is well written"*. BALi’s primary concerns related to theoretical aspects, which we addressed in the rebuttal. BALi confirmed that the questions had been resolved and subsequently increased the score.

Reviewer **N2ms** praised the paper’s *"notable theoretical contribution"* and *"significant practical implications"*, remarking that UFT *"could simplify the fine-tuning strategy"* and that our theory *"formally confirm an intuition about RL in long-horizon tasks"*. N2ms raised questions on assumptions for theory and ablations on experiments, which we addressed later. N2ms maintained **Accept** after the discussion.

Thanks once again for your time, effort, and thoughtful feedback.

Sincerely,

The Authors

---

### Decision · Program_Chairs · 2025-09-17

**Decision:**

Accept (poster)

**Comment:**

**Summary**

This paper tackles the important challenge of effectively combining SFT and RL in LLM post-training. The core contribution is a new training scheme that applies SFT loss to initial tokens and RL loss to later tokens within a single generation. This transition is managed by an annealing schedule that gradually shifts focus from SFT to RL. The authors provide both empirical results and theoretical justification for their approach.

**Strengths**

The paper’s primary contribution is a principled method for integrating SFT and RL. The proposed method with annealing schedule provides a mechanism for a gradual transition from SFT to RL, addressing a known challenge in the field.

**Weaknesses**

The main weakness of the paper lies in the limited generalizability of its findings.
* **Empirical Scope:** The experimental validation is restricted to smaller-scale models (< 3B) on a narrow range of tasks. This makes it difficult to assess whether the observed benefits will transfer to the larger, more capable models, as performance trends at smaller scales do not always generalize predictably.
* **Theoretical Assumptions:** The practical relevance of the theoretical guarantees is questionable, as some of the underlying assumptions do not align with standard practice. For example, key theorems rely on an extremely small $\beta$ value (lines 266, 516), a condition that is not met in typical LLM training configurations.
* **Problem Setting:** The method assumes that a ground-truth prefix (the initial tokens of a desired response) is available for every prompt to apply the SFT loss (line 181). This is a strong assumption that deviates from the standard RL setting for LLMs.

**Recommendation**

Despite the concerns regarding generalizability, the paper introduces an interesting method for a timely problem. It represents a notable attempt to better integrate SFT and RL, and its findings, while preliminary, could serve as a useful data point for future research in this direction.